# Impacts of air pollutants from rural Chinese households under the rapid residential energy transition

Guofeng Shen [1], Muye Ru[1,2], Wei Du[1], Xi Zhu[1], Qirui Zhong[1], Yilin Chen[1,3], Huizhong Shen [1,3], Xiao Yun[1], Wenjun Meng[1], Junfeng Liu[1], Hefa Cheng[1], Jianying Hu[1], Dabo Guan [4] & Shu Tao [1,5]

Rural residential energy consumption in China is experiencing a rapid transition towards clean energy, nevertheless, solid fuel combustion remains an important emission source. Here we quantitatively evaluate the contribution of rural residential emissions to $PM_{2.5}$ (particulate matter with an aerodynamic diameter less than 2.5 μm) and the impacts on health and climate. The clean energy transitions result in remarkable reductions in the contributions to ambient $PM_{2.5}$, avoiding 130,000 (90,000–160,000) premature deaths associated with $PM_{2.5}$ exposure. The climate forcing associated with this sector declines from $0.057 \pm 0.016$ W/m$^2$ in 1992 to $0.031 \pm 0.008$ W/m$^2$ in 2012. Despite this, the large remaining quantities of solid fuels still contributed $14 \pm 10$ μg/m$^3$ to population-weighted $PM_{2.5}$ in 2012, which comprises $21 \pm 14\%$ of the overall population-weighted $PM_{2.5}$ from all sources. Rural residential emissions affect not only rural but urban air quality, and the impacts are highly seasonal and location dependent.

[1] College of Urban and Environmental Sciences, Laboratory for Earth Surface Processes, Peking University, 100871 Beijing, China. [2] Nicholas school of the Environment, Duke University, Durham, NC 27705, USA. [3] School of Civil and Environmental Engineering, Georgia Institute of Technology, Atlanta, GA 30332, USA. [4] School of International Development, University of East Anglia, Norwich, Norfolk NR4 7TJ, UK. [5] Sino-French Institute for Earth System Science, Peking University, 100871 Beijing, China. Correspondence and requests for materials should be addressed to S.T. (email: taos@pku.edu.cn)

Air pollution is one of the most concerning environmental issues in China, causing more than one million premature deaths each year[1,2]. Research has documented that air pollutants are primarily from power generation, industry, transportation, agriculture, and residential activities[3,4]. Although residential fuel consumption contributes a very small fraction of the total energy use in China, it contributes significantly to pollutant emissions and consequently adverse health and climate impacts[2,5,6]. This is because the emission factors (EFs, quantities of air pollutants emitted per unit of fuel consumed) for extensively used solid fuels in this sector are very high. According to the latest estimation, 27% primary $PM_{2.5}$ (particulate matter with an aerodynamic diameter less than 2.5 μm) and 51% BC (black carbon) emissions in mainland China were from the residential sector in 2014, and nearly 80% was from rural areas[7].

Unfortunately, without enough first-hand data, the important role that residential energy transition can play in mitigating health burdens has hardly been justified. The residential sector has long been overlooked for its contribution to air pollution in China[8,9], likely because the relatively low contribution of the residential sector to total energy use was known, whereas very high EFs are often not aware of. Another obstacle for better understanding is the fact that residential energy consumption and emissions were poorly recorded in comparison with other sectors[10,11], leading to higher uncertainties in emission estimations. Consequently, the residential sector is the "shortest wooden bar" in the overall evaluation of emissions and air pollution.

Recently, a thorough nationwide survey was conducted to collect first-hand data on rural residential energy use from 1992 to 2012 in rural China[11]. A rapid transition of rural residential energy from solid fuels towards cleaner energy was revealed. The study also found that the quantities of biomass fuel use and the energy mix transition have been misestimated to a large extent[11]. The new data from this survey provide us with a unique opportunity to improve our estimation of residential contributions to air pollution and to evaluate the impacts of the residential energy transition on health and climate forcing. In fact, the co-impacts of air pollution and co-benefits of emission reduction on health- and climate-relevant air pollutants are often expected and recognized in literature[5,12–16]. For example, studies quantified premature deaths avoided and the influence on radiative forcing induced by using low sulfur jet and ship fuels[13,14], and co-benefits from increasing household insulation in U.S. households have been previously reported[15].

In this study, we quantitatively distinguish emissions from rural residential sources and all other sectors based on the newly compiled emission inventories of major air pollutants, model the contributions of the rural residential sector to ambient $PM_{2.5}$, and evaluate the co-impacts of the residential energy transition on air pollution-associated health and climate radiative forcing (see Method). The reason for focusing on rural areas is because the majority of residential emissions in China are from rural areas where the detailed residential energy survey was targeted.

## Results and discussion
**Contributions to ambient and population-weighted $PM_{2.5}$.** Due to the rapid transition of cooking and heating energy from solid fuels towards clean fuels and electricity, as well as a rapid urbanization, most air pollutant emissions from the rural residential sector have been reduced over the last two decades; however, this sector remains a significant source of many air pollutants, contributing 39% and 46% of the total emissions of BC and organic carbon (OC) in 2012, respectively[11]. Based on the results of atmospheric chemical transport modelling (see Method), in 2012, the increment contributed by rural residential emissions to the national annual mean ambient $PM_{2.5}$ concentration was $5.4 \pm 6.1$ μg/m³ with high spatial variations. This value corresponds to a relative contribution of one-third ($33 \pm 17\%$) of the total $PM_{2.5}$ concentration in ambient air originating from all anthropogenic emission sources including power generation, industry, and transportation.

Because of uneven population distribution, the population-weighted concentration (PWC) was different from the air quality concentration. The contribution from rural residential emissions to the annual mean PWC was $14 \pm 10$ μg/m³, accounting for $21 \pm 14\%$ of the total PWC from all sources. The national average rural residential source associated PWC (14 μg/m³) was almost three times the average ambient $PM_{2.5}$ concentration attributable to rural residential emissions (5.4 μg/m³). Since the National Grade-I standard[17], equal to the Interim Target-1 level set by the World Health Organization (WHO)[18], for the annual mean ambient $PM_{2.5}$ is 35 μg/m³, the contribution from the rural residential sector cannot be ignored in the overall mitigation strategy for combating severe air pollution.

**Impact on rural and urban air quality.** Although the emissions from this sector occur only in rural areas, their contribution to air pollution is not limited to such areas. According to the model results, the absolute contribution of rural residential emissions to the average ambient $PM_{2.5}$ concentration in cities ($13 \pm 10$ μg/m³) was even higher than that in rural areas ($5.3 \pm 6.0$ μg/m³). This outcome occurred because vast areas in western China with very low population densities are far from high emission regions, whereas most populated cities are located in eastern China and surrounded by highly populated rural villages. On the other hand, there was no significant difference in the PWC between rural ($13 \pm 6$ μg/m³) and urban ($14 \pm 13$ μg/m³) areas, due to higher urban population density. In relative terms, the contributions of rural residential emissions in rural areas, for both air concentration ($33 \pm 17\%$) and PWC ($23 \pm 13\%$), were higher than those in urban areas ($21 \pm 15\%$ and $20 \pm 17\%$, respectively), due to strong emissions from other sectors in urban areas.

**Two-decadal changes in rural residential contributions.** While the contribution of the rural residential sector to air pollution remained significant in 2012, it was even higher prior to 2012. Because of the rapid transition of the rural residential energy mix over the past two decades, from domination by solid fuels to clean fuels and electricity, the contribution of this sector to the ambient air concentration has decreased. Figure 1 shows the decreasing trends of the PWC in both absolute (A) and relative (B) terms for urban, rural, and national averages from 1992 to 2012. In 1992, the relative contributions of the rural residential sector to the overall ambient $PM_{2.5}$ and PWC were as high as $45 \pm 11\%$ and $39 \pm 13\%$, respectively. Over the 20-year period, the national average PWC decreased by ~26% from 18 μg/m³ to 14 μg/m³. The relative decrease of the national average was even more rapid from 39% to 21% because the air pollutant emissions from other sectors increased during this same period[3,4,19]. This increase is particularly true for urban areas after 2002. For example, against the decreasing trend in NOx emissions from the rural residential sector, the total NOx emissions from all sources jumped 316% from 1992 to 2012, primarily driven by the massive industrialization and explosive growth of the passenger car fleet[19].

**Higher contributions in winter in northern China.** In comparison with other major emission sectors including power stations, industry, and transportation, after the rapid transition towards clean fuels and electricity, the relative contribution of the residential sector to the annual mean $PM_{2.5}$ PWC ($21 \pm 14\%$) has been relatively low. However, because of the strong seasonality of

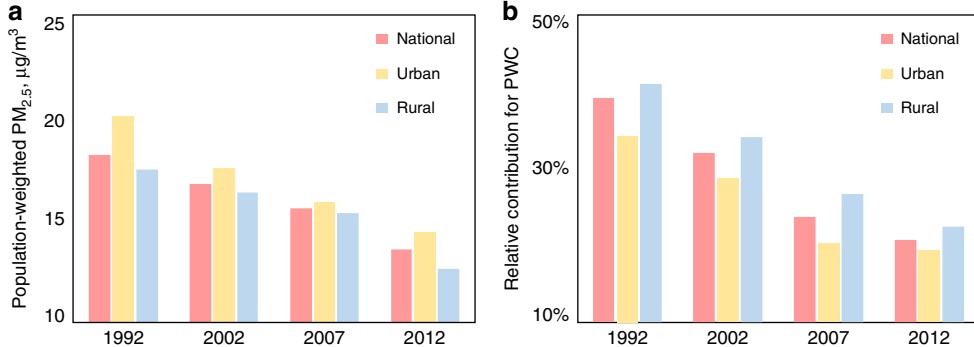

**Fig. 1** Contributions of rural residential emissions to the overall population-weighted ambient PM$_{2.5}$ concentration (PWC) in mainland China from 1992 to 2012. The results are presented as the national total and for the rural and urban areas, in absolute **a** and relative **b** contributions, respectively. Source data are provided as a Source Data file

heating needs in northern China, there has been very high seasonal variation in this contribution. For example, Supplementary Fig. 1 shows the monthly contributions of rural residential emissions to PM$_{2.5}$ PWC in China in 1992 and shows strong seasonality. On a national scale, the rural residential sector contributed 33 μg/m$^3$ PM$_{2.5}$ to ambient air in the three winter months from December to February, while the summer value was only 12 μg/m$^3$.

Although the seasonal patterns were similar among the four study years, the differences between the heating and non-heating seasons increased over this period. The ratios of the relative contributions between winter and summer months increased from 1.8 (1992) to 2.4 (2002), 2.8 (2007), and 3.7 (2012). This trend can be explained by the fact that, at this stage, the rural residential energy transition occurred primarily for cooking activities. The time-sharing fraction of using LPG, biogas, and electricity for cooking increased from 7% in 1992 to 59% in 2012, whereas the fraction for heating only reached 15% in 2012[11]. Because cooking emissions tend to remain constant between seasons, the cooking energy transition-induced decrease in PM$_{2.5}$ concentrations did not vary among months. On the other hand, the heating energy transition, which can lead to seasonal change, occurred at a much slower pace and largely in mid-latitude provinces where heating is needed for no more than a few weeks[11]. In the northern, northwestern, and northeastern regions, where heating is usually needed for several months[20], residents cannot shift to electricity or gas for heating due to the high cost and their low income[11].

**Strong spatial variations with higher contributions in east**. In addition to the temporal changes, there was also strong spatial variation in the contributions of rural residential emissions to ambient PM$_{2.5}$ concentrations, which is shown in Fig. 2a as provincial averages. Although the national average contribution of rural residential sources to the annual mean ambient PM$_{2.5}$ was 5.4 ± 6.1 μg/m$^3$ in 2012, the contributions in eastern China, where the majority of the population reside, were much higher than those in western China. The heating activity not only caused significant seasonal differences (Supplementary Fig. 1) but also different geographical distribution patterns of the contributions, with much higher levels in northern China during heating periods due to high heating demand. The spatial variation was enhanced by population weighting for assessing exposure, as shown by provincial averages in Fig. 2b. The overlay of the sources (emissions) and receptors (population) led to an even sharper contrast between the heavily and less polluted regions.

The current air pollution control strategy focuses on the most polluted and populated regions, including the North China Plain, Guanzhong Plain, and Northeastern Plain, where ambient PM$_{2.5}$ concentrations in winter are generally higher than those in summer[21]. In addition to poor dispersion conditions, heating activity in rural areas is definitely one of the most important reasons for the severe air pollution in winter. Despite increasing concerns on this issue[2], the contribution to exposure has not been well quantified in the literature. To fill this data gap, the contributions of rural residential emissions to the PM$_{2.5}$ PWC in all provinces were calculated. The results are listed in Supplementary Table 1 in both absolute and relative terms. The relatively high standard deviation values indicate high spatial variation. For the provinces in the North China Plain and Guanzhong Plain, the annual mean contributions were not higher than the national averages. However, the winter (January) values were much higher. For most provinces in the northeast, north, and northwest, except for Beijing and Tianjin, the relative contributions in January were between 27% and 51%, accounting for approximately one-third to one-half of the PWC in the region. The values for the residential emissions alone are already approaching or exceeding the National Grade-I standard of 35 μg/m$^3$[17]. More importantly, the absolute contributions of rural residential emissions to the PWC in cities in eastern China are often close to, or even higher than, those in rural areas. For example, the relative contributions of the rural residential sector to PM$_{2.5}$ PWC in urban and rural areas in Heilongjiang were 24 ± 14% and 32 ± 10%, respectively. The values were 41 ± 29% and 18 ± 9% for the urban and rural areas in Inner Mongolia, respectively, likely due to the very low population density in rural areas. Although the relative contributions in western provinces such as Xingjiang (34 ± 25%) and Qinghai (23 ± 14%) were even higher, the absolute contributions were much lower (2.9 ± 2.8 and 3.3 ± 2.6 μg/m$^3$, respectively) than those in the east. The contributions were much higher in 1992 compared to those in 2012, indicating a rapid reduction due to the clean energy transition. For example, the absolute and relative contributions in January in 1992 were as high as 53 ± 25 μg/m$^3$ and 60 ± 15% in Hebei, and 77 ± 16 μg/m$^3$ and 69 ± 7% in Henan provinces, respectively, which are among the most populated provinces in China. The rural residential energy transition resulted in a significantly reduced contribution to overall air pollution. Although this trend is promising, the rural residential sector still plays a critical role in PM$_{2.5}$ pollution, especially in northern China during winter. The emissions from this source

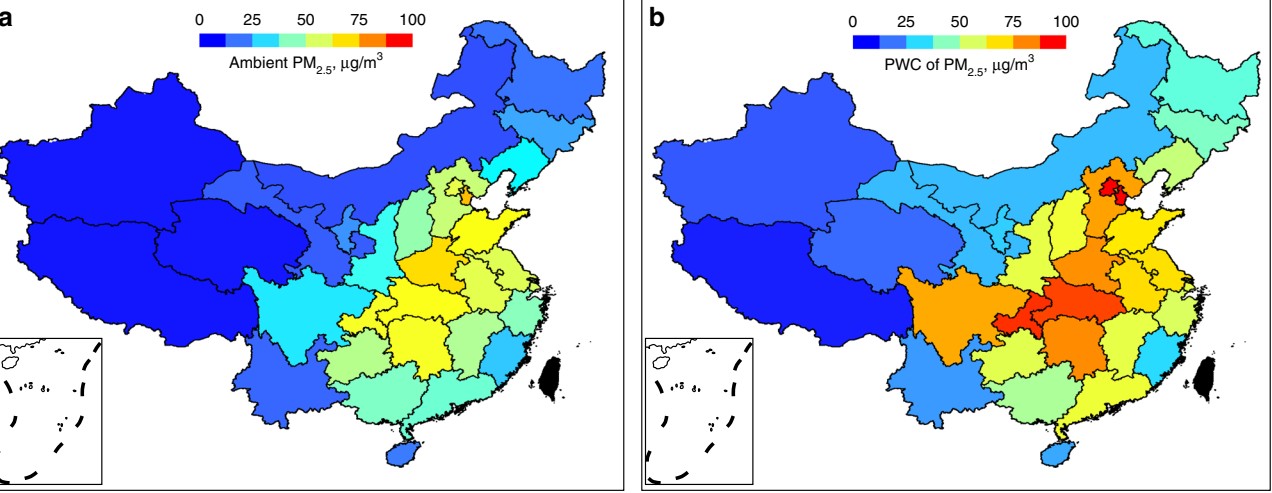

**Fig. 2** Spatial distribution of rural residential emission contributions in 2012 in mainland China. Results are shown as absolute contributions to provincial mean air quality concentrations **a** and population-weighted PM$_{2.5}$ concentrations **b**. Source data are provided as a Source Data file

should be remediated before the nationwide air pollution issue can finally be solved.

**Co-benefits of rural residential energy transition.** With significant contributions to exposure, adverse health effects from rural residential emissions are inevitable. Additionally, the residential energy transition away from solid fuels can reduce health impacts. Based on the modelled PWC and the corresponding dose-response relationship, the annual premature deaths caused by ambient PM$_{2.5}$ originating from rural residential emissions were calculated (see Method) for each year studied, and the result is shown in Fig. 3. The health impacts of other pollutants, such as O$_3$ and CO were not evaluated in this study due to the relatively low impacts originating from this sector and a shortage of emission data.

This study found that the rural residential emission-associated premature deaths were as high as 570,000 (370,000–710,000 with the 95% uncertainty interval) in 1992 and decreased to 210,000 (140,000–260,000) in 2012 according to the GEMM model. Although baseline mortality rates changed rapidly during this period, the decrease in the mortality attributable to rural residential emissions was still significant from 1992 to 2012, with the influence of the population growth, aging, and age-standardised mortality rate adjusted (Supplementary Fig. 2). The estimated number of deaths avoided due to decreased rural residential exposure was approximately 130,000 (90,000–160,000). During the same period, because of general increase in emissions from other anthropogenic sources, the total premature deaths induced by exposure to PM$_{2.5}$ from all sources increased by 15%, which is consistent with the trend in the GBD study[1]. This outcome indicates the important health consequences of this emission source as well as the significant health benefits of the rural energy transition. Compared with all other emission sources, the relative contribution of residential emissions to the total premature deaths caused by exposure to ambient PM$_{2.5}$ from all sources decreased from 33% in 1992 to 10% in 2012 (Supplementary Fig. 3). This rapid decrease was due to decreased emissions from this sector whereas increased emissions from many other sources. The difference between the relative contributions to PM$_{2.5}$ concentrations (21%) and to health effects (10%) in 2012 is partly due to the nonlinearity of the dose-response curves. As shown in Fig. 4, because of the increased emissions from other sources from 1992 to 2012, the frequency

distribution of PM$_{2.5}$ PWC shifted towards the right, whereas the response curves were relatively flat.

Sector emission-associated premature deaths have been investigated previously[2,5,6,22,23]. The differences among these studies are expected and can be explained by the differences in inventories, models, and methods adopted in health impact evaluation. Although Chafe et al. (2014)[6] reported relatively small numbers of premature deaths in China (220,000, 170,000, and 130,000 in 1990, 2005, and 2010, respectively), because only cooking fuels were considered, a similar decreasing trend was demonstrated. Two other studies focused on the health impacts of sector emissions on a global scale and reported in 2005 that a total of 207,000 deaths in China could be avoided if global residential emissions were zeroed[22], and that 26,000 deaths could be avoided if emissions from this sector were reduced by 20% in 2010[23]. Lelieveld et al. (2015)[2] reported that a total of 434,000 premature deaths, which was ~32% in terms of the relative contribution, were attributable to PM$_{2.5}$ and O$_3$ originating from residential emissions in 2010 in China. Considering that the contribution from residential sources to O$_3$ precursors was much smaller than that from other sectors[24], and that residential emissions in China were predominantly from rural areas[3,25], this value is higher than our estimate. This difference can be partially explained by the difference in emission inventories. Our inventory was characterized by a rapid increase in clean energy from 1992 to 2012[11], a trend very different from that of previous data[26,27].

Supplementary Fig. 4 compares the geographical distributions of premature deaths originating from rural residential emissions between 1992 and 2012. In both years, the higher mortality areas were the North China Plain, Guanzhong Plain, and the Sichuan Basin. The mortalities decreased throughout all of eastern China over the 20-year period, with an increasing downward trend towards the coastal areas. This trend is likely linked to faster socioeconomic development and the consequent rapid rural residential energy transition in the east. Supplementary Fig. 4 also shows the cumulative frequency distributions of the calculated grid mortality in 1992 and 2012. The fraction of the population facing a risk above 10$^{-3}$ due to PM$_{2.5}$ from all emission sources increased from 76% in 1992 to 84% in 2012; however, in this period, the relative contribution of rural residential emissions to premature deaths shifted in the opposite direction from 33% to 10%.

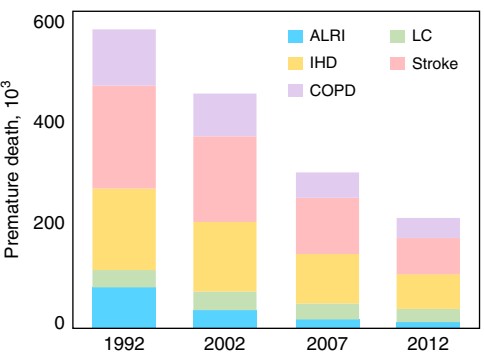

**Fig. 3** Model-calculated population risk associated with exposure to PM$_{2.5}$ originating from rural residential emissions from 1992 to 2012. The results are shown for chronic obstructive pulmonary disease (COPD), cerebrovascular disease (stroke), ischaemic heart disease (IHD), lung cancer (LC), and acute lower respiratory infections (ALRIs). Source data are provided as a Source Data file

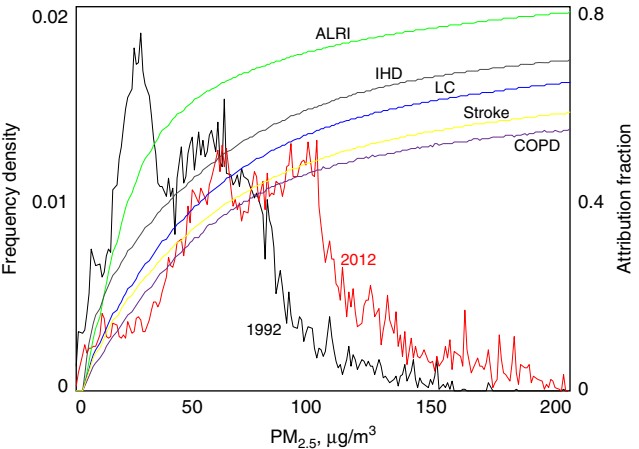

**Fig. 4** Dose-response relationship and the cumulative frequency distributions of the total PWC from all sources in 1992 and 2012. The results are shown for chronic obstructive pulmonary disease (COPD), cerebrovascular disease (stroke), ischaemic heart disease (IHD), lung cancer (LC), and acute lower respiratory infections (ALRIs). Source data are provided as a Source Data file

The annual mean radiative forcing values attributable to rural residential emissions in China were calculated for the four study years as the differences between the radiative forcing derived based on all anthropogenic emissions and that based on all but rural residential emissions using the OSCAR model (see Method). Among the components studied, CO$_2$ and BC contributed positively to climate forcing, whereas primary organic aerosol (POA), nitrate, and sulfate contributed negatively. According to our results, five components from the rural residential sector contributed $+0.006 \pm 0.004$ (CO$_2$), $+0.049 \pm 0.007$ (BC), $-0.019 \pm 0.002$ (POA), $-0.002 \pm 0.000$ (nitrate), and $-0.003 \pm 0.000$ (sulfate) W/m$^2$ in 2012, with a net contribution of $+0.031 \pm 0.008$ W/m$^2$ (Fig. 5). This contribution is equivalent to 12.5% of the total forcing caused by CO$_2$ from all sources in 2012. Of the five components, BC was the most important component, whereas the CO$_2$ contribution was much smaller. Indeed, CO$_2$ from the rural residential sector was merely 2.6% of the total CO$_2$ from all sources in 2012[28,29], mainly because the majority of biomass fuels, including all crop residues and most fuelwood consumed in rural

China, are carbon neutral[30]. Meanwhile, fossil fuels consumed in the rural residential sector accounted for a very small fraction of the total[4]. From 1992 to 2012, the rural residential energy transition from solid fuels to clean fuels resulted in a reduction in climate forcing. The absolute contributions of all five components, whether positive or negative forcers, decreased over the two decades. Although the CO$_2$ emissions from LPG and electricity consumption increased, the decrease in CO$_2$ emissions from fuelwood use was faster, leading to a net reduction of CO$_2$ emissions from this sector from 1992 to 2012. Because BC contributed much more than all others in absolute terms, despite the opposite effects of other forcing components, the residential energy transition-induced decrease in the BC emissions from biomass burning led to a significant decrease in net radiative forcing of 61%, from $0.057 \pm 0.016$ W/m$^2$ to $0.031 \pm 0.008$ W/m$^2$ during this 20-year period.

One limitation of this study is that only the influences on ambient air pollution and associated health impacts were addressed. In reality, another equally, if not more, important effect caused by solid fuel consumption, and impacted by the transition away from it, is household air pollution associated with emissions from indoor solid fuel combustion[31,32]. Both ambient and household air quality can benefit from the rural residential energy transition. Unfortunately, mainly due to the lack of quantitative data on parameters including fugitive emissions, air exchange rate, indoor circulation, etc., household exposure is subject to limitations and constraints at this time. These constraints and limitations are borne by almost all existing literature on the health impacts of indoor air pollution, including the latest GBD study[33]. There is an urgent need to develop a reliable method to characterize the process so that the overall exposure and health impacts can be fully addressed. The health benefits of the transition are expected to be even greater when household air pollution is considered. According to the evidence provided in this study, it is recommended that replacing the remaining solid fuels with affordable cleaner fuels or electricity in the rural residential sector be incorporated into the overall air pollution mitigation strategy in China. Similarly, this substitution should be encouraged in other developing countries in South Asia, Africa, and South America.

In summary, a rapid transition of rural residential energy away from solid fuels from 1992 to 2012 resulted in significantly reduced contributions of this sector to ambient PM$_{2.5}$ concentrations, subsequently reducing both adverse health impacts and climate forcing. Despite these changes, solid fuels are still used extensively in rural China, particularly for heating, contributing significantly to air pollution and population exposure.

## Method

**Emission data**. The emission inventories of CO$_2$ and major air pollutants, including SO$_2$, CO, PM$_{2.5}$, PM$_{10}$, BC, OC, NOx, and NH$_3$, from rural residential and other sources with $0.1° \times 0.1°$ spatial and daily temporal resolutions for 1992, 2002, 2007, and 2012 were sourced from http://inventory.pku.edu.cn, which was updated recently to include the latest information on the rural residential energy transition, population growth, and urbanization[11]. Air pollutant emissions for the rural residential sector were mainly based on two recently updated databases. A new energy database was established based on a rural residential energy survey that covered detailed energy use activities, including staple food cooking, side dish preparation, water boiling, animal feed heating, and space heating for more than 34,400 households and daily biomass fuel consumption quantities for more than 1600 households[11]. The rural residential energy database covered detailed information on various energy types including coal, honeycomb briquette, straw, corncob, fuelwood, brushwood, charcoal, LPG, biogas, and electricity. The emission factor database was re-compiled based on carefully screened data from published studies conducted in China. For CO$_2$ emission calculation, all crop residues were considered carbon neutral, and for fuelwood, the non-renewable fractions for all individual provinces in China were sourced from the literature[30,34]. The VOC emissions were obtained from the EDGAR-HTAP dataset[35,36].

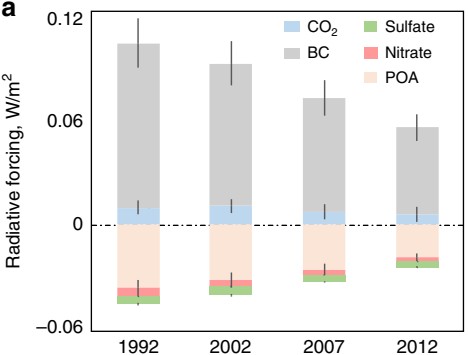
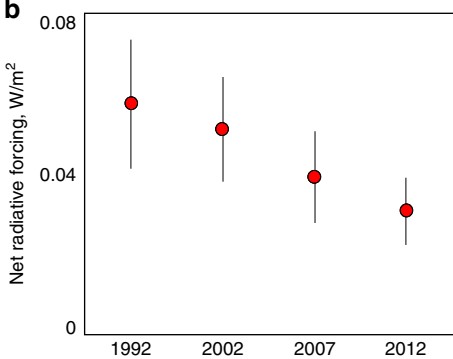

**Fig. 5** Radiative forcing associated with emissions from rural residential sector. **a** Radiative forcing caused by $CO_2$, BC, POA, sulfate, and nitrate originating from rural residential emissions in 1992, 2002, 2007, and 2012. **b** The net forcing of these components. Error bars indicate standard deviations generated from the Monte Carlo simulation. Source data are provided as a Source Data file

**Atmospheric chemical transport modelling**. In this study, WRF/Chem version 3.5 was applied to model the daily $PM_{2.5}$ concentrations in ambient air for 1992, 2002, 2007, and 2012 in China (13°N–56°N; 67°E–143°E, Supplementary Fig. 5). In brief, the RADM2 chemical mechanism for the gas phase and the MADE/SORGAM aerosol scheme were adopted with a $50 \times 50$ km$^2$ horizontal resolution and a 5-min time step. Because population density varies with short distances, the calculated $PM_{2.5}$ concentrations were downscaled to 30-arc sec in both longitude and latitude, based on wind fields and the emissions inventory[37]. In brief, the model- calculated concentrations were interpolated to a finer resolution using 0.1 degree emission inventory as a proxy with the wind-field justified. The detailed downscaling method can be found elsewhere[38]. Another commonly used method for downscaling is land-use regression, in which land-use is primarily used as a proxy for emissions[39]. It is reasonable to expect that direct use of emissions in this study can provide better results. A statistically significant relationship between emissions and annual mean $PM_{2.5}$ concentrations in individual grids has been demonstrated[40]. The WRF meteorological inputs were evaluated against observations from China Earth International Exchange Stations. The normalized mean bias (NMB) and normalized mean error (NME) were calculated for all validations. The NMB for surface pressure, temperature, relative humidity, and wind speed were −3%, −7% ~ −10%, −2% ~ −9 and 32% ~43%, respectively (Supplementary Table 2). According to the results of a reduced-form modelling, the meteorological conditions in these 4 years were close to the multi-year average[40]. Prior to 2013, there was no official routine monitoring program for ambient $PM_{2.5}$ in China[41,42], and sampled $PM_{2.5}$ data were only available from several field studies reported in the literature and those released from the U.S. Embassy. To validate our modelling results, in addition to comparing with available observational data[41–44] (Supplementary Figs. 6 and 7), the modelled $PM_{2.5}$ concentrations were also compared to those retrieved from the satellite remote sensing, which have been widely used in the scientific community[45–47], and compared to those estimated based on visibility records in China[48]. Overall, our model simulated concentrations were slightly higher than the observation data (NMB = 25–30%, Supplementary Fig. 6 and NMB = 3%, Supplementary Fig. 7), but considerably lower than those retrieved from the satellite and visibility retrieved results (NMB = −61% ~ −5%, Supplementary Table 3). Supplementary Figure 8 shows the spatial pattern of the differences between the modelled $PM_{2.5}$ concentrations in this study and those retrieved from the satellite-derived data[45]. Our results were lower in the sparsely populated western area and higher in some southeast sites. Previous studies found that satellite-derived $PM_{2.5}$ might underestimate $PM_{2.5}$ levels during heavy pollution episodes and overestimate $PM_{2.5}$ concentrations in low pollution cases[41,49,50]. The discrepancy is also likely from uncertainties in chemical transport model inputs such as air emissions of $PM_{2.5}$ precursors, and missing mechanisms in the model[51–53]. We further compared the modelled major $PM_{2.5}$ components with measured data, including black carbon (NMB = 67%), organic carbon (NMB = −3%), sulfate (NMB = −12%), nitrate (NMB = 49%), and ammonium (NMB = 38%) (Supplementary Fig. 9). The differences could come from the uncertainties in modelling, as well as uncertainties in limited observations with different sampling and analytical methods. With all the results taken into consideration, the model-predicted ambient $PM_{2.5}$ concentrations and the spatiotemporal variations are generally acceptable.

To quantitatively evaluate the contributions from rural residential sources, a normalized marginal method (Supplementary Note 1)[54,55] was adopted by running the model three times using emission scenarios of: a 20% reduction in the rural residential sources; a 20% reduction in all but the residential sources; and all sources. The influences on the rural and urban areas were evaluated separately. Rural and urban areas were spatially distinguished by using a dynamic urban mask developed Shen et al., (2017)[38]. In brief, the urban mask was developed by extracting built-up areas (urban) using remote sensing data, night-time light data, and population census data, based on county-specific thresholds.

**Health and climate impact assessment**. Population exposure to ambient $PM_{2.5}$ in China in 1992, 2002, 2007, and 2012 was derived based on the modelled near-surface $PM_{2.5}$ concentrations and the population densities[38]. The PWC of a region with more than one grid cell was calculated as $\Sigma(P_i \cdot C_i)/P_t$, where $P_i$ and $C_i$ are population and air $PM_{2.5}$ concentration at the $i^{th}$ grid, and $P_t$ is the total population in the domain of interest. The change in rural population density due to rapid urbanization in China was considered[38]. Premature deaths from acute lower respiratory infections (ALRIs) for children under five, ischemic heart disease (IHD), cerebrovascular disease (Stroke), chronic obstructive pulmonary disease (COPD), and lung cancer (LC) were estimated from the exposure and the latest hazard ratios in the Global Exposure Mortality Model (GEMM) constructed by Burnett et al. (2018)[56]. The population-attributable fraction and annual premature death rate were calculated based on the provincial background disease burdens reported by the Global Burden of Disease (GBD) study[57]. Changes in the background disease burden over the 20-year period were considered, and the ratios of urban and rural background disease incidents to total provincial background disease incidents were derived[58]. The relative contributions of exposure, population growth, population ageing, and baseline mortality rates were analysed after Cohen et al., (2017)[1]. Health impact models, especially does-response functions, affect estimated premature deaths considerably. For comparison, premature deaths were also calculated using the Integrated Risk Function developed by the GBD[1] and the results are compared in the Supplementary Table 4. As expected, the GEMM model yielded more premature deaths than the GBD model. However, the relative contributions of rural residential emissions to the total premature deaths and the temporal trend were similar between these two approaches (Supplementary Figs. 3).

The annual mean radiative forcing attributable to rural residential emissions was the difference between two scenarios of all anthropogenic emissions with and without rural residential sources. To estimate the radiative forcing, the global biogeochemical cycle model OSCAR (v2.1) was used[28]. The OSCAR model is a reduced-form model used widely in the climate change research community. This model was developed with the three principles of embedding as many components and processes as possible, building as a meta-model capable of emulating the sensitivity of models with higher resolution or superior complexity, and comprising as a dynamic model of the Earth system[29]. The model outputs include radiative forcing of each climate-relevant component and global surface temperature change, and the inputs are emissions of various compounds and drivers related to land-use and land-cover change. A detailed description of the model was presented by Gasser et al. (2017)[29]. To estimate the contributions from rural residential sources, the model was run three times using different emission scenarios, in line with those model scenarios in the atmospheric chemical transport modelling, and the normalized marginal method was used (Supplementary Note 1). The derived climate forcing components discussed in this study included $CO_2$, BC, POA, sulfate, and nitrate, of which residential coal and biomass burning are usually the larger emitters of these compounds or their precursors. The contributions of residential combustion emissions to ozone precursors and methane are relatively small compared to its contribution to aerosol[24,59]. The OSCAR model has been widely used in the climate change research community[28]. Although this is a simple box-model without seasonal variation, it is very efficient in terms of computation. As a parametric model with most parameters previously calibrated against complex models during a meta-modelling process[29], OSCAR fits our purpose well.

**Uncertainty analysis**. A Monte Carlo simulation (1000 times) was conducted to characterize the uncertainty of the estimated health impacts. The results reflect the influences of both the exposure level and the dose-response relationship including slope and nonlinearity. Because of the heavy computation load, a Monte Carlo simulation could not be conducted for the transport modelling. According to the

results from a previous research the variation of $PM_{2.5}$ concentrations is similar to that of primary $PM_{2.5}$ emissions[40], which was close to 10% in China[3]. Therefore, for each grid, a 10% uncertainty in air $PM_{2.5}$ was assumed and used in the uncertainty analysis for the health impact assessment. The sensitivity analysis (Supplementary Fig. 10) showed that differences in estimated premature deaths with the assumed grid $PM_{2.5}$ concentration uncertainties at 10%, 15%, and 25% were within 1%, indicating that the overall uncertainty in health outcomes was affected largely by dose-response functions compared with the uncertainty in $PM_{2.5}$ concentration[60]. The uncertainties of estimated health impacts were calculated based on the given distribution of parameters from the GEMM[56], by applying a 1000-run Monte Carlo simulation. The variations derived from the Monte Carlo simulation are strongly associated with the nonlinearity of the dose-response relationship. The importance of the dose-response relationship for uncertainty has been previously reported[60]. The results are presented at a 95% uncertainty interval (which ranges between the 2.5th and 97.5th percentiles).

## Data availability

The source data underlying Figs. 1–5, Supplementary Figs. 1-4 and Supplementary Figs 6-10 and Supplementary Tables 1-4 are provided as a Source Data file.

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

## Acknowledgements

This work is funded by the National Natural Science Foundation of China (Grants 41830641, 41571130010, 41821005, and 41629101), the 111 program (B14001), and the Undergraduate Student Research Training Program of the Ministry of Education. The authors thank Prof. J. West (University of North Carolina, Chapel Hill) for kindly providing the sectoral contribution data for a comparison, Prof. Z. Ma (Nanjing University) and Prof. R. Martin (Dalhousie University) for providing the satellite-based PM$_{2.5}$ concentration in China.

## Author contributions

S.T. coordinated and supervised the project, S.T. and G.S. designed the present experiment and modeling analysis, G.S., M.R., W.D., H.C. and S.T. prepared and interpreted survey data, X.Z., Q.Z., Y.C., H.S., X.Y., W.M. and J. L. performed the model analysis and validation. X. Z., Q.Z. and X. Y. completed all figures. G.S., J.L., H.C., J.H., D.G. and S.T. analyzed the simulation results. S.T. and G.S. drafted and revised the manuscript, and all coauthors contributed to the interpretation of the results and to the text. All authors read the manuscript and approved the submission.

## Additional information

**Competing interests:** The authors declare no competing interests.

