## [Peer Review File · Nature Communications]

Reviewers' comments:

Reviewer #1 (Remarks to the Author):

This manuscript quantifies the contributions of rural residential emission to PM2.5 and the impacts on health and climate. Although the title connects Chinese rural residential emissions with residential energy transitions, the changes in the emissions can be attributed also to other important factors, such as the urbanization induced population migration (Shen et al., 2017). In addition, each component involved in the assessment (e.g. contribution of rural households emissions, health impacts, and climate impacts) is highly uncertain. Specific comments are listed below:

- (1) The authors fail to show that the rural households emissions are reliable;
- (2) climate forcing inferred from models highly depends on the capability of model in capturing aerosol components. It would be great if the authors can validate it across China;
- (3) uncertainty analysis was conducted for health impacts, yet such uncertainty can not represent the real case. GBD updates IER coefficient every year, and the differences of those IER coefficients for each update are way higher than the uncertainty shown in this study. It is not convincing that the 95% interval can represent the uncertainty of health impacts;
- (4) The health impacts evaluated are in terms of outdoor pollution, but rural households emissions should have way larger health impacts in terms of indoor exposure. With energy transition, the major health benefits should come from indoor not outdoor, as people stay more hours indoor everyday and are exposed to solid fuel burning directly. Please refer to many papers by Prof. Kirk Smith.

Shen, H., Tao, S., Chen, Y., Ciais, P., Güneralp, B., Ru, M., Zhong, Q., Yun, X., Zhu, X., Huang, T., Tao, W., Chen, Y., Li, B., Wang, X., Liu, W., Liu, J., Zhao, S., Urbanization-induced population migration has reduced ambient PM2.5 concentrations in China. *Science Advances* 2017, 3(7): e1700300.

Reviewer #3 (Remarks to the Author):

Air Pollution from Rural Chinese Households under Fast Residential Energy Transition

Summary

This study presents air quality, health and climate impacts from residential combustion emissions over a two decade period from 1992 to 2012, when China underwent a fast residential energy transition. Using a model-based framework, the authors provide analyses to quantify the AQ and health benefits, and focus on identifying rural vs. urban and summer vs. winter season impacts. Given the growing health concerns of air pollution, this study is a topic of interest, and could provide valuable information for scientists and policy-makers.

Major Comments

I find there is inconsistency in how the results are presented, which often leads to confusion on what is the actual metric being focused on. For e.g., the WHO standard for PM_{2.5} is an annual standard at 35 micrograms/m³. There is confusion in the units when micrograms/m³ and milligrams/m³ are used somewhat interchangeably more than once. Given the large shift in implications depending on the actual units, and the potential for incorrect units being used in the health risk and climate analyses, I find it hard to justify recommending the paper be accepted as is. Suggest the authors go through this very carefully, and improve the clarity of what metrics are being presented.

Minor Comments

Lines 23 – 52: In describing the background for this study, I am sure there are other studies that have tried to assess air quality, health and climate impacts, using similar or different frameworks. The background literature review is a bit weak from that perspective. There are couple of global or regional studies that come to mind. See Barrett et al, EST 2012 on a global study of jet emissions, and Levy et al, ERL 2016 on a U.S. based study on residential combustion emissions. Suggest the authors expand their literature review to include additional citations, and more importantly comment on the methods they used vs. others in the past for similar assessments.

L30: Change “climate-relevant compounds” to “climate-relevant pollutants”

L50: “diameters” -> “diameter”

L58: Move “respectively” to end of sentence

L61: Why is the deviation (6.1) higher than the mean (5.4)? Please explain what this means for the underlying dataset. And again, how does this translate to 33±17% of the total?

L61, L67, L73: Are these annual averages, or some other temporal period? Since the WHO standard of 35 $\mu\text{g}/\text{m}^3$ is annual, how are these daily averages from the model computed? Is this for a specific day, or an average of multiple days, or something else? Please clarify

L61: Here, the rural contribution is stated as 5.4 ± 6.1 . On L79, it is stated as 5.3 ± 6.0 . Which is correct?

L72: Please spell out WHO, and provide reference to WHO guidance

L82: Please define "exposure concentration", and why are its units ($\mu\text{g}/\text{m}^3$) same as air concentrations?

L131: Which month are these daily contributions presented for? How do these numbers relative to the previously reported values (in temporal scales)?

L242: What is the significance of $10^{-3.5}$?

L274-275: What does "equal concentrations in outdoor and indoor" mean? Are you suggesting that there is 100% infiltration from outdoor to indoor, and hence the indoor environment is in equilibrium with the outdoor?

L283: "south America -> "South America"

L286: Please explicitly call out the list of pollutants.

L287 and L292: The 3rd year of interest is stated as 2007 for emissions and 2009 for WRF-Chem. Please clarify

L294: Please state that the "RADM2' chemical mechanism refers only to the gas-phase chemical mechanism

L295 – 297: “The calculated concentrations of primary PM_{2.5} were downscaled to 30” spatially, based on wind fields and the emissions inventory.” What is the intent of this downscaling? And please clarify the units of 30”. Is this really 30 inches? And how do you downscale from an areal extent (50x50 sq km) to a linear distance (30”)?

L297: Were the met fields also evaluated? What are the main findings there?

L303: Please provide more info about the “proportional method”

L321 – 323: But previously in L301-303, , it was stated that the emissions were reduced by 20% in scenarios 1 and 2? Is there a consistency issue here?

L331: please provide justification for 10% COV used here

L333: What are the 4 parameters and their values? Please provide additional info.

L463 Fig 1: How do you fit a curve from 4 points? Please clarify interpolation or fitting done to these points to generate these curves. Are the units for PM_{2.5} really in mg/m³?

L470 Fig 2: Why such high values for rural contribution in left figure?

L474 Fig 3: Units are problematic – ambient in mg/m³, exposure in µg/m³? Please clarify. All narrative in main body used only µg/m³. Also, what do these ranges mean? Was some transformation done to the data? Suggest using easy to interpret ranges, unless there is a need to transform them.

L478 Fig 4: It seems that there is a step-function like decrease at around the year 2005 and 2009 in the left panel. What explains this?

Supplemental Info:

Fig S3 Panel A and B: Legend label – please fix font issue with right most text. Is it 0.5 or something else? Also, what are the 2 inset panels? They are very fuzzy and not adding any value?

Fig S3 Panel C: Y-axis label seems problematic. Can you also comment on what these distributions imply?

Fig S5: Are the concentration units really in mg/m³? Please provide justification for why these 2 cities were chosen to be plotted? What are the findings? Can you explain the fairly high overpredictions of the model during Spring and Summer months in Beijing? Further, please provide quantitative measures of model performance, instead of timeseries of observed and modeled values.

Fig S6: Please provide additional information on how the normalizing was done? Usually, after normalizing, you have a max scale of 1. So, what do these maxima of 3.8 (for obs) and 3.1 (for mod) mean, and what are the implications of having -ve values in normalized concentrations? Further, what is meant by “geospatial variation”?

Table S1: Several values (both absolute and relative) have SD > mean. Please comment on this.

Reviewer #4 (Remarks to the Author):

This is an interesting paper that quantifies the contributions of emissions from rural Chinese households on air pollution, air pollution deaths, and radiative climate forcing. Air pollution in China is clearly important, and the authors suggest that this sector is important, even though its importance has decreased through time. The authors seem to have done a good job of quantifying uncertainty.

Overall, I have several questions and concerns over methods and results that are not sufficiently clear to me. Once these concerns are addressed, I am hopeful that this paper will be a good contribution to the literature with some results that may be policy-relevant for China but perhaps also elsewhere. However, it is not immediately apparent to me that a revised paper would be sufficiently interesting to warrant publication in a high profile journal like Nature Communications.

In particular, there have been several global studies of the contributions of different emissions sectors to air pollution deaths. The authors reference just one of these studies and show that their results are significantly different (smaller) – this is an interesting finding that deserves more exploration – but this suggests that the current paper is not extremely novel. One could consider, for example, if the authors repeated this analysis to highlight other sectors – the electricity generation sector, industry, transportation – would each of those studies merit publication in Nature Communications? If not, then what is the motivation for publishing this? The authors have more work to do, in my opinion, to justify why a focus on residential emissions should be published in a top journal, especially when they show that the contribution of residential emissions to the total air pollution is decreasing.

General concerns:

1) The focus is on “rural Chinese households” or “rural residential”. Why is the focus only on residential emissions from rural regions? Why not include residential emissions from urban areas also? Or it seems that the focus is on solid fuel combustion, which is likely to be mostly in rural areas. But I thought that solid fuels are used for home heating and cooking in some urban areas, or if that is not common now, it would have been common in 1992 which is also modeled here. Why are urban emissions excluded here? And is it just solid fuels only, or does it include other residential fuels? More clarity is needed to better explain the focus, with some discussion to justify that choice.

2) The first two sentences of the abstract seem to contradict one another, and this contradiction shows up in many places in the paper. Residential emissions are a “missing piece of the puzzle” suggesting that they are not considered in air pollution control. But at the same time, there has been a remarkable energy transition that has reduced emissions substantially and has made the relative contribution of this sector decrease through time. It seems to me that language like “missing piece of the puzzle” is overselling the importance of this work.

3) When a conversion is made from solid fuels to LPG or electricity, emissions are reduced, but they do not go to zero. It is not clear to me whether these emissions are included in the analysis of changes in residential emissions, but I would guess that they are not. More clarity is needed on what is modeled, and if these increased emissions from LPG and electricity are neglected then some discussion of this point is needed.

4) In addition to comparing results of the number of deaths with Lelieveld et al (ref. 7), the results should be compared with others in the literature – Silva et al (2016, EHP), Chafe (2014, EHP), Liang (2018, ACP). The authors found a relatively smaller contribution of residential than Lelieveld, but it is not clear to me that this is a fair comparison because of definitions of “residential” – whereas

Lelieveld considers all residential emissions, the focus here seems to be on rural emissions and on solid fuel (or are other rural residential emissions included?).

Related to this, the smaller contribution compared with Lelieveld is interesting and suggests more analysis and discussion. The authors say that Lelieveld's emissions (from commonly used global datasets) are too high because they don't adequately reflect the energy transition which is captured here. Here the authors implicitly seem to think that their inventory is better than the global inventories, but they have not justified that conclusion.

5) The atmospheric modeling does not seem to be new here, but is the results of references 28 and 29. Or if new modeling was done in this study (such as for the earlier years), that is not entirely clear. So this might reduce the novelty of this paper to the health and climate impacts.

6) I don't know what the "exposure concentrations" mean. It says it is population-weighted, but does that mean conc x population in each grid cell? The units of Figure 3 bottom row show values between 10^{-5} and 10^{-10} , with units of $\mu\text{g m}^{-3}$, which would not seem to make sense with that definition. Should the units be "people $\mu\text{g m}^{-3}$ "?

7) The methods of assessing radiative forcing are entirely unclear to me. A radiative forcing is a difference between two states – it is not clear what the years in figure 5 are evaluated relative to (no residential emissions)? Does Figure 5 show annual global average forcing? In a single year from emissions changes, or in some future year representing steady state? Is the forcing accumulated over many years? I had thought that the authors might use the results of the WRF-chem simulations to drive regional radiative forcing estimates, but that does not appear to be what is done here – rather a biogeochemical model and a "normalized marginal method" is used. I am not familiar with the marginal method, although it may be sufficient for this purpose. I don't understand what the biogeochemical model is used for – perhaps something with the carbon cycle. Critical here is what is assumed for CO₂ – whether the biomass is sustainably harvested or not – and that assumption is not at all clear. In line 256 it says that most fuel wood is carbon neutral – is that what is assumed in this study? In lines 260-262, it says that increase LPG and electricity consumption were compensated for by a decrease in fuel wood consumption – so do the results show both changes, and was this also done for the health calculations? Why are only these species evaluated and not, for example, the emissions of ozone precursors such as CO and NO_x?

8) The model evaluation given (Figs. S5 and S6) is pretty bare bones with no statistical summaries. Could the authors model 2015 for better comparison with the observations that were available in 2015 than in 2012?

9) The study ends without conclusions – the last paragraph is almost entirely about the fact that household exposure is not addressed here.

10) The paper seems long to me for a Nature journal. There is a lot of long discussion of results that I don't think are very hard to understand (the difference between absolute and relative results), repetition of results (different in winter), and statements of policy relevance scattered throughout, so I have the sense that the whole paper could be conveyed in much fewer words if needed. But if the length is acceptable with the journal, then I have no problem.

More specific comments:

l. 10 – I think that the deaths should be “per year”. This needs to be clear throughout the paper.

l. 11 – a 20 year period – when?

l. 40-46 – This is good, but it has not been shown that the China-based inventory is more accurate than the international ones. I think it is likely better, but it has not been demonstrated to be more accurate, and no further argument is given that it is better.

l. 66 – The authors should more carefully define “ambient” and “population-weighted”. I don't think ambient is usually used in this way.

l. 124 – The discussion around Fig. 2 is based on this graph of only one city in one month. It leaves the question of whether this is representative of other locations and months. The authors have the data to analyze much more than this.

l. 151-153 – This suggests that coal and biomass were used very recently in urban areas, bringing into question why this isn't modeled.

l. 156 – I don't understand the text.

l. 208 – This is not how the word co-benefits is usually used. The authors should define what is a co-benefit of what. And this section doesn't really evaluate benefits – rather it evaluates trends through time.

l. 292 – I think 2009 should be 2007.

l. 297-299 – Rather than just saying that it was evaluated against measurements, it would be better to summarize the main findings of that comparison.

Fig. 2 – When is this shown? What is the x-axis?

Fig. 3 – A logarithmic scale may be appropriate here, but I find $10^{1.6}$ to be not very useful on the color scale – better to show the actual numbers. I don't understand what is plotted in the bottom row or its units.

Fig. 4 – in panel a, does the blue at the top signify anything? In panel b, what is “the attributed factor” (is this the “attributable fraction”)? Why does 1992 look like lower concentrations than 2012, when other graphs show decreases through time?

Reviewer-1

All line numbers referred to in these responses are from the previous version. – authors.

Comment

This manuscript quantifies the contributions of rural residential emission to PM_{2.5} and the impacts on health and climate. Although the title connects Chinese rural residential emissions with residential energy transitions, the changes in the emissions can be attributed also to other important factors, such as the urbanization induced population migration (Shen et al., 2017). In addition, each component involved in the assessment (e.g. contribution of rural households emissions, health impacts, and climate impacts) is highly uncertain. Shen, H., Tao, S., Chen, Y., Ciais, P., Güneralp, B., Ru, M., Zhong, Q., Yun, X., Zhu, X., Huang, T., Tao, W., Chen, Y., Li, B., Wang, X., Liu, W., Liu, J., Zhao, S., Urbanization-induced population migration has reduced ambient PM_{2.5} concentrations in China. *Science Advances* 2017, 3(7): e1700300.

Response

Influences of urbanization on rural (but not urban) populations, energy, emissions, and exposure were considered in the study. The sentence “Change in rural population density due to rapid urbanization in China was considered. (Shen et al., 2017).” was added after “... and the population densities.” in line 309. The phrase “as well as rapid urbanization” was inserted into the first sentence of the **Results and Discussion** (line 56). The first sentence of **Emission data in Methodology** was also revised to cover the information: “The emission inventories of CO₂ and major air pollutants, including SO₂, CO, PM_{2.5}, PM₁₀, TSP, BC, OC, NO_x and NH₃, from rural residential and other sources with 0.1° × 0.1° spatial and daily temporal resolutions for 1992, 2002, 2007, and 2012 were sourced from <http://inventory.pku.edu.cn>, which was updated recently to include the latest information on the rural residential energy transition, population growth, and urbanization.”.

The results are indeed associated with high uncertainty, which is addressed by the uncertainty analysis (refer to the **Uncertainty Analysis** section in **Methodology**) and model validation (lines 297-299), both of which were further revised based on specific comments from all reviewers. For detailed revisions, please refer to the responses below.

Limitations of the approach associated with the uncertainty are discussed in the last paragraph of the **Results and Discussion**. Detailed responses to specific comments can be found below.

Specific comments are listed below:

Comment

1) The authors fail to show that the rural households emissions are reliable;

Response

We did not provide detailed information on how the rural household emissions were derived in this manuscript because details have been presented in a paper published previously (Tao et al., 2018). In brief, we conducted a nationwide survey in 2012 to collect detailed residential energy information in rural China. In addition to the 2012 survey, data were also recalled for 1992, 2002, and 2007. Instead of only recording primary energy, detailed time-sharing data were collected for five activities including staple food cooking, side dish preparation, water boiling, animal feed heating, and space heating. In addition, the total quantities of coal, LPG, honeycomb briquette, and electricity used were surveyed. Valid data for over 34,400 households were gathered, covering more than 98% of municipalities in mainland China. For biomass fuels, another campaign was conducted to weigh the daily consumption of biomass fuels for 1,640 households, which were combined with time-sharing data to generate total quantities of biomass fuel use. These data provide first-hand information on rural residential energy consumption and transition. The emissions, which are also included in the previously published paper, were derived based on these data along with a large emission factor database compiled over multiple years.

Although it is difficult, but not impossible, to evaluate the reliability of the emission estimation, the emission data used in this study were based on the best data available. The following sentence was added to the first paragraph of **Methodology** to address this issue: “Air pollutant emissions for the rural residential sector were mainly based on two recently updated databases. A new energy database was established and mainly based on a rural residential energy survey that covered detailed energy use activities, including staple food cooking, side dish preparation, water boiling, animal feed heating, and space heating for more

than 34,400 households and daily biomass fuel consumption quantities for more than 1,600 households (Tao et al., 2018). The rural residential energy database covered detailed information on various energy types including coal, honeycomb briquette, straw, corncob, fuelwood, brushwood, charcoal, LPG, biogas, and electricity. The emission factor database was re-compiled based on carefully screened data from published studies conducted in China.”.

Reference

Tao S, et al., Quantifying the rural residential energy transition in China from 1992 to 2012 through a representative national survey. *Nature Energy* 3, 567-573 (2018).

Comment

2) climate forcing inferred from models highly depends on the capability of model in capturing aerosol components. It would be great if the authors can validate it across China;

Response

To validate the model-calculated concentrations of major PM_{2.5} components, data were collected from the literature for comparison. In addition to black carbon and organic carbon, the total concentration of sulfate, nitrate, and ammonium was used as an indicator of secondary aerosols.

A sentence was added to line 299: “The model was also validated for major PM_{2.5} components, including black carbon, organic carbon, and SNA (sum of sulfate, nitrate, and ammonium). The results are shown in Fig. S7.”. Additionally, a figure (Fig. S7) was added to the supplementary information along with a list of literature.

Fig. S7 Comparison of the calculated concentrations of major PM_{2.5} components including black carbon (A), organic carbon (B) and SNA (sum of sulfate, nitrate, and ammonium) (C) with those observed. The dashed and dotted lines represent the error ranges of two and five times, respectively. Normalized mean bias (NMB) and normalized mean error (NME) are also shown. Source data are provided as a Source Data file.

Comment

3) uncertainty analysis was conducted for health impacts, yet such uncertainty can not represent the real case. GBD updates IER coefficient every year, and the differences of those IER coefficients for each update are way higher than the uncertainty shown in this study. It is not convincing that the 95% interval can represent the uncertainty of health impacts;

Response

With the recently released Global Exposure Mortality Model (GEMM) now available (Burnett et al., 2018), health impact was re-calculated for the revised version, and all results were updated accordingly. In addition, the uncertainty analysis was re-conducted with more reasonable assumptions considering both exposure calculation and concentration-response functions. The resulting uncertainty ranges are much larger than previous ones. Based on the new results, the following revisions were made:

Lines 9-11: The phrase was revised to “The associated PM_{2.5} exposure-induced premature deaths per year decreased 52% from 490,000 (310,000-620,000 as the 95% uncertainty interval) to 230,000 (160,000-290,000) from 1992 to 2012.”.

Line 213-215: The sentence was revised to “This study found that the rural residential emission-associated premature deaths were as high as 490,000 (310,000-620,000 within the 95% uncertainty interval) in 1992 and decreased by more than 50% to 230,000 (160,000-290,000) in 2012.”.

Line 218-220: The sentence was revised to “...the relative contribution of residential emissions to the total premature deaths caused by exposure to ambient PM_{2.5} from all sources decreased from 34% in 1992 to 10% in 2012”.

Line 220-230: The sentences (“It was previously reported that 32% ... due to the nonlinearity of the dose-response curves”) were rewritten as follows: “Sector emission associated premature deaths have been investigated previously (Butt et al., 2016; Chafe et al., 2014; Silva et al., 2016; Lelieveld et al., 2015; Liang et al., 2018). The differences among these studies are expected and can be explained by the differences in inventories, models, and methods adopted in health impact evaluation. Although Chafe et al. (2014) reported relatively small numbers of premature deaths in China (220,000, 170,000, and 130,000 in 1990, 2005 and 2010, respectively), because only cooking fuels were considered, a similar decreasing trend was demonstrated. Two other studies focused on the health impact of sector emissions on a global scale and reported in 2005 that a total of 207,000 deaths in China could be avoided if global residential emissions were zeroed (Silva, et al., 2016), and that 26,000 deaths could be avoided if emissions from this sector were reduced by 20% in 2010 (Liang et al., 2018). Lelieveld et al. (2015) reported that a total of 434,000 premature deaths, which was approximately 32% in terms of relative contribution, were attributable to PM_{2.5} and O₃ originating from residential emissions in 2010 in China. Considering that the contribution from residential sources to O₃ precursors was much smaller than that from other sectors (Li et al., 2017) and that residential emissions in China were predominantly from rural areas (Huang et al., 2014; Zhu et al., 2018), this value is higher than our estimate. This difference can be partially explained by the difference in emission inventories. Our inventory was characterized by a rapid increase in clean energy from 1992 to 2012 (Tao et al., 2018), a trend very different to that of previous data.”.

Lines 240-244: The sentences were revised to “The fraction of the population facing a risk above 10⁻³ due to PM_{2.5} from all emission sources increased from 66% in 1992 to 90% in 2012. However, the relative contribution of rural residential emissions to premature deaths shifted in the opposite direction from 34% in 1992 to 10% in 2012.”.

Lines 310-313: The sentence was revised to: “Premature deaths from acute lower respiratory infections (ALRIs) for children under five, ischemic heart disease (IHD), cerebrovascular disease (stroke), chronic obstructive pulmonary disease (COPD), and lung cancer (LC) were estimated from the model-calculated exposure and the latest hazard ratios in the Global Exposure Mortality Model (GEMM) constructed by Burnett et al. (2018).”.

Lines 332-336: The sentence was revised to: “The uncertainties of estimated health impacts were calculated based on the given distribution of parameters from GEMM (Burnett et al., 2018), by applying a 1,000-run Monte Carlo simulation.”.

Fig. 4 and Fig. S3 were redrawn based on the new results

Fig. 4 Model-calculated population risk associated with exposure to PM_{2.5} originating from rural residential emissions from 1992 to 2012 (A); and B) the nonlinear dose-response relationship and the cumulative frequency distributions of the total PWC from all sources in 1992 and 2012. The results are shown for chronic obstructive pulmonary disease (COPD), cerebrovascular disease (stroke), ischaemic heart disease (IHD), lung cancer (LC), and acute lower respiratory infections (ALRIs). Source data are provided as a Source Data file. Source data are provided as a Source Data file.

Fig. S3 Spatial distribution of log-scale premature deaths induced by ambient PM_{2.5} originating from rural residential emissions in China in 1992 (A) and 2012 (B). The five PM_{2.5} exposure-associated diseases include acute lower respiratory infections (ALRIs), ischemic heart disease (IHD), cerebrovascular disease (Stroke), chronic obstructive pulmonary disease (COPD), and lung cancer (LC). The cumulative frequency distributions of the grid mortality induced by PM_{2.5} from all sources (blue lines) and all but the rural residential source (red lines) for 1992 (solid lines) and 2012 (dash lines) and the cumulative frequency distributions of mortality rate in 1992 and 2012 are also shown (C). The differences between the solid and dash lines had shrunk from 1992 (solid) to 2012 (dash), indicating reduction of relative contributions of rural residential sources. Source data are provided as a Source Data file.

References

- Butt, E.W. et al. The impact of residential combustion emissions on atmospheric aerosol, human health, and climate. *Atmos. Chem. Phys.* 16, 873-905 (2016).
- Burnett, R., et al. Global estimates of mortality associated with long-term exposure to outdoor fine particulate matter. *PNAS* 115: 9592-9597 (2018).
- Chafe, Z. et al. Household cooking with solid fuels contributes to ambient PM_{2.5} air pollution and the burden of disease. *Environ. Health Perspect.* 122, 1314-1320 (2014).
- Huang, Y., et al., Quantification of global primary emissions of PM_{2.5}, PM₁₀ and TSP from combustion and industrial process sources. *Environ. Sci. Technol.* 48, 13834-13843 (2014).
- Lelieveld, J., Evans, J.S., Fnais, M., Giannadaki, D. & Pozzer, A. The contribution of outdoor air pollution sources to premature mortality on a global scale. *Nature* 525, 367-371 (2015).
- Li, M., et al., MIX a mosaic Asian anthropogenic emission inventory under the international collaboration framework of the MICS-Asia and HTAP. *Atmos. Chem. Phys.* 17, 935-963 (2017)
- Silva, R. A., Adelman, Z., Fry, M., & West, J. J. The impact of individual anthropogenic emissions sectors on the global burden of human mortality due to ambient air pollution. *Environ. Health Perspect.* 124, 1776-1784 (2016)
- Tao, S. et al. Quantifying the Rural Residential Energy Transition in China from 1992 to 2012 through a Representative National Survey. *Nature Energy*, 3, 567-573 (2018).
- Liang, C. et al. HTAP2 multi-model estimates of premature human mortality due to intercontinental transport of air pollution and emission sectors. *Atmos. Chem. Phys.* 18, 10497-10520 (2018).

Comment

4) The health impacts evaluated are in terms of outdoor pollution, but rural households emissions should have way larger health impacts in terms of indoor exposure. With energy transition, the major health benefits should come from indoor not outdoor, as people stay more hours indoor everyday and are exposed to solid fuel burning directly. Please refer to many papers by Prof. Kirk Smith.

Response

We are aware of this limitation, which is addressed in the second to last paragraph (revised version) of the **Results and Discussion**. With currently available data, the common way to quantify indoor exposure is based on the average indoor air concentrations of households using specific fuel types. Indeed, such an approach has been used by us (Chen et al., 2018) and others (Mestl et al., 2007). However, this method is associated with much higher uncertainty compared with ambient air quality modelling, primarily due to very limited field observations and large variation of indoor air concentrations. We think this uncertainty would be substantially reduced by using a process model with stove fugitive emissions incorporated. Unfortunately, such a technique is not yet available under real world emission conditions. Until this technique is fully developed and applied to collect enough field data, it is not sufficient to bring the

uncertainty of indoor air quality estimation down to a level similar to that of ambient air quality modelling. The difference in uncertainty levels between ambient air quality modelling and indoor air quality modelling is the reason that we have chosen not to include indoor exposure in the current study.

References

- Chen, Y., et al. Estimating household air pollution exposures and health impacts from space heating in rural China. *Environ. International* 119, 117-124 (2018).
- Mestl, H.E.S., Aunan, K., Seip, H., Wang, S., Zhao, Y., & Zhang, D. Urban and rural exposure to indoor air pollution from domestic biomass and coal burning across China. *Sci. Total Environ.* 377, 12-26 (2007).

Reviewer-3

All line numbers referred to in these responses are from the previous version. – authors.

Comment

This study presents air quality, health and climate impacts from residential combustion emissions over a two decade period from 1992 to 2012, when China underwent a fast residential energy transition. Using a model-based framework, the authors provide analyses to quantify the AQ and health benefits, and focus on identifying rural vs. urban and summer vs. winter season impacts. Given the growing health concerns of air pollution, this study is a topic of interest, and could provide valuable information for scientists and policy-makers.

Response

Thank you very much for the comments, which have been valuable for improving our manuscript.

Major Comments**Comment**

I find there is inconsistency in how the results are presented, which often leads to confusion on what is the actual metric being focused on. For e.g., the WHO standard for PM_{2.5} is an annual standard at 35 micrograms/m³. There is confusion in the units when micrograms/m³ and milligrams/m³ are used somewhat interchangeably more than once. Given the large shift in implications depending on the actual units, and the potential for incorrect units being used in the health risk and climate analyses, I find it hard to justify recommending the paper be accepted as is. Suggest the authors go through this very carefully, and improve the clarity of what metrics are being presented.

Response

We apologize for the errors. The units were checked and “ $\mu\text{g}/\text{m}^3$ ” was used for PM_{2.5} concentration throughout the revised manuscript.

Minor Comments**Comment**

Lines 23 – 52: In describing the background for this study, I am sure there are other studies that have tried to assess air quality, health and climate impacts, using similar or different frameworks. The background literature review is a bit weak from that perspective. There are couple of global or regional studies that come to mind. See Barrett et al, EST 2012 on a global study of jet emissions, and Levy et al, ERL 2016 on a U.S. based study on residential combustion emissions. Suggest the authors expand their literature review to include additional citations, and more importantly comment on the methods they used vs. others in the past for similar assessments.

Response

The entire introduction, except for the last paragraph (study objectives), was rewritten to briefly include more relevant studies. Additionally, the previous introduction text was shortened by reducing the discussion on energy transition. The rewritten induction (the first three paragraphs) is as follows:

“Research has documented that residential solid fuel use in China emits large quantities of air pollutants (Shen et al., 2013; Huang et al., 2014; Wang et al., 2013; Liu et al., 2016), leading to adverse health impacts. For example, an estimated 32% of the premature mortality induced by exposure to ambient air pollution in China in 2010 could be attributed to residential emissions (Lelieveld et al., 2015). The result of another modelling study suggested that 220,000 premature deaths were associated with solid fuels used for cooking in China in 1990, and this number decreased to 130,000 in 2010 (Chafe et al., 2014). A similar estimation also showed that 256,000 premature deaths per year were associated with this sector in India (Conibear et al., 2018). In addition to health impacts, the emission of primary and secondary aerosols, especially black carbon (BC) from residential solid fuel consumption, also has important climate implications (Wang et al., 2012; Butt et al., 2016).

With challenges from both the health and climate fronts, there is growing interest in pollution co-impacts and emission reduction co-benefits, which can be quantified by integrating transport, exposure, and impact

modelling. For example, one study estimated that using ultra-low sulfur jet fuel could prevent 2,300 premature deaths annually and influence radiative forcing by +3.3 mW/m² (Barrett et al., 2012). Similarly, other research reported that using low-sulfur fuels could reduce premature deaths and radiative cooling from ship-related aerosols by 34% and 80%, respectively (Sofiev et al., 2018). In another study, over 308,000 premature adult deaths and a direct radiative effect between -66 mW/m² and +21 mW/m² could be attributed to global residential emissions in 2000 (Butt et al., 2016). Additional research reported that annual CO₂ emissions and many air pollutants from electricity generating units and direct residential combustion could be substantially reduced by increasing insulation in all single-family homes in the United States, resulting in a co-benefit of approximately 320 deaths prevented per year (Levy et al., 2016).

Recently, a nationwide survey was conducted to collect first-hand data of rural residential energy use from 1992 to 2012 in China (Tao et al., 2018). The results determined that solid fuel use for cooking and heating had been quickly replaced with liquefied petroleum gas (LPG), pipeline natural gas (PNG), biogas, and electricity, primarily due to rapid socioeconomic development. Similar rates have been reported by other surveys (Xie et al., 2014; Chen et al., 2016; Qiu et al., 2015). One direct result may be significant emission reductions of major air pollutants from this sector (Tao et al., 2018). Nevertheless, solid fuels remain as major energy types, especially for heating, leading to high emissions. In comparison, the data from the International Energy Agency and the Food and Agriculture Organization recorded a mere 10% reduction in biomass fuel consumption over this 20-year period, resulting in very different results for solid fuel use and trends (IEA, 2017; FAO, 2017).”.

References

- Shen, H. et al. Global atmospheric emissions of polycyclic aromatic hydrocarbons from 1960 to 2008 and future predictions. *Environ. Sci. Technol.* 47, 6415-6424 (2013).
- Huang, Y. et al. Quantification of global primary emissions of PM_{2.5}, PM₁₀, and TSP from combustion and industrial process sources. *Environ. Sci. Technol.* 48, 13834-13843 (2014).
- Wang, R. et al. High-resolution mapping of combustion processes and implications for CO₂ emissions. *Atmos. Chem. Phys.* 13, 5189-5203 (2013).
- Liu, J. et al. Air pollutant emissions from Chinese households: a major and underappreciated ambient pollution source. *Proc. Natl. Acad. Sci. USA* 113, 7756-7761 (2016).
- Lelieveld, J., Evans, J.S., Fnais, M., Giannadaki, D. & Pozzer, A. The contribution of outdoor air pollution sources to premature mortality on a global scale. *Nature* 525, 367-371 (2015).
- Chafe, Z. et al. Household cooking with solid fuels contributes to ambient PM_{2.5} air pollution and the burden of disease. *Environ. Health Perspect.* 122, 1314-1320 (2014).
- Conibear, L. et al. Residential energy use emissions dominate health impacts from exposure to ambient particulate matter in India. *Nature Commun.* 9, 167 (2018).
- Wang, R. et al. Black carbon emissions in China from 1949 to 2050. *Environ. Sci. Technol.* 46, 7595-7603 (2012).
- Butt, E. W. et al. The impact of residential combustion emissions on atmospheric aerosol, human health, and climate. *Atmos. Chem. Phys.* 16, 873 (2016).
- Barrett, S. R. et al. Public health, climate, and economic impacts of desulfurizing jet fuel. *Environ. Sci. Technol.* 46, 4275 (2012).
- Sofiev, M. et al. Cleaner fuels for ships provide public health benefits with climate tradeoffs. *Nature Commun.* 9, 406 (2018).
- Levy, J. I. et al. Carbon reductions and health co-benefits from US residential energy efficiency measures. *Environ. Res. Letters* 11, 034017 (2016).
- Tao, S. et al. Quantifying the Rural Residential Energy Transition in China from 1992 to 2012 through a Representative National Survey. *Nature Energy*, 3, 567-573 (2018).
- Xie, Y. & Hu, J.W. An introduction to the China Family Panel Studies (CFPS). *Chinese Soc. Rev.* 47, 3-29 (2014).
- Chen, Y. L. et al. Transition of household cookfuels in China from 2010 to 2012. *Appl. Energ.* 184, 800-809 (2016).
- Qiu, H. G. et al. Renewable energy consumption in rural China: current situation and major driven factors. *J. Beijing Inst. Technol.* 17, 10-15 (2015).
- International Energy Agency. World Energy Statistics and Balances http://data.iaea.org/ieastore/product.asp?dept_id=101&pf_id=205. Accessed on Dec. 18, 2017.
- Food and Agriculture Organization of the United Nations (FAO) (2017) FAOSTAT: <http://www.fao.org/faostat/en/#data>. Accessed on Jan. 1, 2018.

Comment

L30: Change “climate-relevant compounds” to “climate-relevant pollutants”

Response

The term has not been used in the revised text any more (refer to the above response to the previous comment).

Comment

L50: “diameters” -> “diameter”

Response

Changed accordingly.

Comment

L58: Move “respectively” to end of sentence

Response

Moved accordingly.

Comment

L61: Why is the deviation (6.1) higher than the mean (5.4)? Please explain what this means for the underlying dataset. And again, how does this translate to $33\pm 17\%$ of the total?

Response

The relatively large standard deviation (6.1) compared with the mean (5.4) indicates very high variation, especially spatial variation. The following phrase was added: “, indicating high spatial variation”.

The value of $33\pm 17\%$ was not derived directly from the value of $5.4\pm 6.1 \mu\text{g}/\text{m}^3$. Instead, this value was derived from the calculated contributions of the model results with rural residential sources, all but rural residential sources, and all sources.

Comment

L61, L67, L73: Are these annual averages, or some other temporal period? Since the WHO standard of $35 \mu\text{g}/\text{m}^3$ is annual, how are these daily averages from the model computed? Is this for a specific day, or an average of multiple days, or something else? Please clarify

Response

These figures are all annual averages. The word “average” on line 61 was replaced with “annual mean”, and the phrase “annual mean” was also inserted on lines 67 and 72, accordingly.

Comment

L61: Here, the rural contribution is stated as 5.4 ± 6.1 . On L79, it is stated as 5.3 ± 6.0 . Which is correct?

Response

This figure is correct. The two numbers are contributions to the entire country and to only rural areas in mainland China, respectively.

Comment

L72: Please spell out WHO, and provide reference to WHO guidance

Response

This text was revised accordingly, and a reference was added.

References

World Health Organization. WHO Air quality guidelines for particulate matter, ozone, nitrogen dioxide and sulfur dioxide. Global update 2005. Summary of risk assessment. Geneva, Switzerland. 2006.

Comment

L82: Please define “exposure concentration”, and why are its units ($\mu\text{g}/\text{m}^3$) same as air concentrations?

Response

The term “exposure concentration” was replaced with PWC (population-weighted concentration), which first appears on lines 65-66. The sentence “the contribution to $\text{PM}_{2.5}$ exposure concentration (population-weighted) was different from that to ambient air concentration.” was replaced with “the population-weighted concentration (PWC) was different from the model-calculated concentration.”, and the term “exposure concentrations” in the following text was replaced with “PWC” throughout the manuscript.

Comment

L131: Which month are these daily contributions presented for? How do these numbers relative to the previously reported values (in temporal scales)?

Response

The date was January as indicated on line 125.

Comment

L242: What is the significance of $10^{-3.5}$?

Response

This text was replaced by “ 10^{-3} ”. Correspondingly, the percentages shown on lines 243-244 have been updated. Specifically, “88%” was replaced with “66%”, and “92%” was replaced with “90%”.

Comment

L274-275: What does “equal concentrations in outdoor and indoor” mean? Are you suggesting that there is 100% infiltration from outdoor to indoor, and hence the indoor environment is in equilibrium with the outdoor?

Response

It is a common practice to quantify health impact based only on ambient air concentrations, primarily because indoor air quality is difficult to model at a regional scale. This statement was not correct, and the sentence was deleted accordingly.

Comment

L283: “south America -> “South America”

Response

Revised accordingly.

Comment

L286: Please explicitly call out the list of pollutants.

Response

The following phrase was inserted: “including SO₂, CO, PM_{2.5}, PM₁₀, TSP, BC, OC, NO_x, and NH₃”.

Comment

L287 and L292: The 3rd year of interest is stated as 2007 for emissions and 2009 for WRF-Chem. Please clarify

Response

It should be 2007 for both emission and air quality modelling. Apologies for the mistake, which has been corrected.

Comment

L294: Please state that the “RADM2’ chemical mechanism refers only to the gas-phase chemical mechanism

Response

The phrase “for the gas phase” was added, accordingly.

Comment

L295 – 297: “The calculated concentrations of primary PM_{2.5} were downscaled to 30” spatially, based on wind fields and the emissions inventory.” What is the intent of this downscaling? And please clarify the units of 30”. Is this really 30 inches? And how do you downscale from an areal extent (50x50 sq km) to a linear distance (30”)?

Response

A phrase was added at the beginning of the sentence to address the purpose: “Because population density varies with short distances ...”.

“30” spatially” was revised to “30-arc sec in both longitude and latitude”, and another sentence was added at the end of the sentence: “This detailed method can be found in the literature (Shen et al., 2017).”.

References

Shen, H.Z. et al. Urbanization-induced population migration has reduced ambient PM_{2.5} concentrations in China. *Sci. Adv.* 3, e1700300 (2017).

Comment

L297: Were the met fields also evaluated? What are the main findings there?

Response

We did not perform any evaluation on MET fields. Here, a sentence was added to the revised version to address this concern: “Satisfactory results for simulating meteorological fields in China with the WRF model have been reported previously (Zhao et al., 2012; Wang et al., 2016).”.

Reference:

Zhao, P., Wang, J., Xia, J., Dai, Y., Sheng, Y., & Yue, J. Performance evaluation and accuracy enhancement of a day-ahead wind power forecasting system in China. *Renewable Energy* 43, 234-241 (2012).

Wang, L., Zhang, Y., Wang, K., Zheng, B., Zhang, Q., & Wei, W. Application of Weather Research and Forecasting Model with Chemistry (WRF/Chem) over northern China: Sensitivity study, comparative evaluation, and policy implications. *Atmos. Environ.* 124, 337-350 (2016).

Comment

L303: Please provide more info about the “proportional method”

Response

The following phrase was added to the end of the sentence: “with relative contributions of various sectors unchanged after the standardization.”.

Comment

L321 – 323: But previously in L301-303, it was stated that the emissions were reduced by 20% in scenarios 1 and 2? Is there a consistency issue here?

Response

This was a 20% reduction. The sentences were rewritten as: “... three times using emission scenarios of 1) rural residential sources reduced by 20%; 2) all but the rural residential sources reduced by 20%;...”.

Comment

L331: please provide justification for 10% COV used here

Response

The sentences were revised as follows: “Because of the heavy computation load, the Monte Carlo simulation could not be conducted for the transport modelling. According to the results from previous study, the variation of PM^{2.5} concentrations is similar to that of primary PM_{2.5} emissions (Zhong et al., 2018), which was close to 10% in China (Huang et al., 2014). Therefore, a 10% coefficient of variation was assumed ...”.

References

Huang, Y. et al. Quantification of global primary emissions of PM_{2.5}, PM₁₀, and TSP from combustion and industrial process sources. *Environ. Sci. Technol.* 48, 13834-13843 (2014).
Zhong, Q., et al. Distinguishing emission-associated ambient air PM_{2.5} concentrations and meteorological factor-induced fluctuations. *Environ. Sci. Technol.* 52, 10416-10425 (2018).

Comment

L333: What are the 4 parameters and their values? Please provide additional info.

Response

Those are four parameters in the GBD IER model. This sentence was deleted in the revised version based on the comment of another reviewer.

Comment

L463 Fig 1: How do you fit a curve from 4 points? Please clarify interpolation or fitting done to these points to generate these curves. Are the units for PM_{2.5} really in mg/m³?

Response

The curves were replaced with bars. The figure caption was revised accordingly. The unit “mg” was a mistake and was corrected.

Comment

L470 Fig 2: Why such high values for rural contribution in left figure?

Response

This was not correctly labelled, and the mistake was corrected accordingly.
The relative contribution was between 30-45% (right y-axis). This is a wintertime case in Hebei where the large consumption of solid fuels for space heating resulted in high emissions and contributions to ambient PM_{2.5}.

Comment

L474 Fig 3: Units are problematic – ambient in mg/m³, exposure in µg/m³? Please clarify. All narrative in main body used only µg/m³. Also, what do these ranges mean? Was some transformation done to the data? Suggest using easy to interpret ranges, unless there is a need to transform them.

Response

This should be µg/m³, and the mistake was corrected. The labels were revised to a simple log-scale accordingly. Only two maps of the annual mean ambient PM_{2.5} concentrations and population-weighted PM_{2.5} concentrations are used in the revised version.

Fig. 3 Spatial distribution of annual mean model calculated $PM_{2.5}$ concentrations (A) and population-weighted $PM_{2.5}$ concentrations (B) attributed to rural residential emissions in 2012 in China. Source data are provided as a Source Data file.

Comment

L478 Fig 4: It seems that there is a step-function like decrease at around the year 2005 and 2009 in the left panel. What explains this?

Response

Apologies as the x-axis was not correctly marked. 2002 should be in the middle of the axis. To avoid confusion, this map was replaced with a bar chart instead of an area chart. The decrease accelerated during the period from 2002 to 2007 and slowed down after 2007. This mistake was corrected.

Fig. 4 Model-calculated population risk associated with exposure to $PM_{2.5}$ originating from rural residential emissions from 1992 to 2012 (A); and B) the nonlinear dose-response relationship and the cumulative frequency distributions of the total PWC from all sources in 1992 and 2012. The results are shown for chronic obstructive pulmonary disease (COPD), cerebrovascular disease (stroke), ischaemic heart disease (IHD), lung cancer (LC), and acute lower respiratory infections (ALRIs). Source data are provided as a Source Data file. Source data are provided as a Source Data file.

Supplemental Info:

Comment

Fig S3 Panel A and B: Legend label – please fix font issue with right most text. Is it 0.5 or something else? Also, what are the 2 inset panels? They are very fuzzy and not adding any value?

Response

This figure legend was revised for clarification. The inserted panels are part of the background maps.

Fig. S3 Spatial distribution of log-scale premature deaths induced by ambient PM_{2.5} originating from rural residential emissions in China in 1992 (A) and 2012 (B). The five PM_{2.5} exposure-associated diseases include acute lower respiratory infections (ALRIs), ischemic heart disease (IHD), cerebrovascular disease (Stroke), chronic obstructive pulmonary disease (COPD), and lung cancer (LC). The cumulative frequency distributions of the grid mortality induced by PM_{2.5} from all sources (blue lines) and all but the rural residential source (red lines) for 1992 (solid lines) and 2012 (dash lines) and the cumulative frequency distributions of mortality rate in 1992 and 2012 are also shown (C). The differences between the solid and dash lines had shrunk from 1992 (solid) to 2012 (dash), indicating reduction of relative contributions of rural residential sources. Source data are provided as a Source Data file.

Comment

Fig S3 Panel C: Y-axis label seems problematic. Can you also comment on what these distributions imply?

Response

This label was fixed. This is the cumulative frequency distribution of grid mortalities under different scenarios.

The following sentence was added to the end of the Fig. S3 caption: “The differences between the solid and dash lines had shrunk from 1992 (solid) to 2012 (dash), indicating reduction of relative contributions of rural residential sources.”

Comment

Fig S5: Are the concentration units really in mg/m³? Please provide justification for why these 2 cities were chosen to be plotted? What are the findings? Can you explain the fairly high over predictions of the model during Spring and Summer months in Beijing? Further, please provide quantitative measures of model performance, instead of time series of observed and modeled values.

Response

The unit should be µg/m³, and the mistake was corrected accordingly.

At that time, these were the only time series data available (the US Embassy in Beijing and the Consulate in Shanghai). A sentence was added to the figure caption: “In 2012, these were the only available time-series data in China.”

Unfortunately, we are not able to explain the over predictions (Shanghai).

Normalized mean bias (NMB) and normalized mean error (NME) were calculated and shown in the figure.

The following sentence was added to address the validation finding: “Normalized mean bias (NMB) and normalized mean error (NME) were calculated for all validations. The model calculations generally agree with the observations in all validation cases.” on line 299.

Comment

Fig S6: Please provide additional information on how the normalizing was done? Usually, after normalizing, you have a max scale of 1. So, what do these maxima of 3.8 (for obs) and 3.1 (for mod) mean, and what are the implications of having -ve values in normalized concentrations? Further, what is meant by “geospatial variation”?

Response

It should be standardization instead of normalization. This mistake was corrected by replacing “normalized” with “standardized” here and on line 304. Additionally, “geospatial variation” was revised as “spatial variation”.

Comment

Table S1: Several values (both absolute and relative) have $SD > mean$. Please comment on this.

Response

The following sentence was added after “... means in both absolute and relative terms” on line 184: “The relatively high values of standard deviations indicate high spatial variation.”

Reviewer-4

All line numbers referred to in these responses are from the previous version. – authors.

Comment

This is an interesting paper that quantifies the contributions of emissions from rural Chinese households on air pollution, air pollution deaths, and radiative climate forcing. Air pollution in China is clearly important, and the authors suggest that this sector is important, even though its importance has decreased through time. The authors seem to have done a good job of quantifying uncertainty.

Response

Thank you very much for reviewing and commenting on the manuscript.

Comment

Overall, I have several questions and concerns over methods and results that are not sufficiently clear to me. Once these concerns are addressed, I am hopeful that this paper will be a good contribution to the literature with some results that may be policy-relevant for China but perhaps also elsewhere. However, it is not immediately apparent to me that a revised paper would be sufficiently interesting to warrant publication in a high profile journal like Nature Communications. In particular, there have been several global studies of the contributions of different emissions sectors to air pollution deaths. The authors reference just one of these studies and show that their results are significantly different (smaller) – this is an interesting finding that deserves more exploration – but this suggests that the current paper is not extremely novel. One could consider, for example, if the authors repeated this analysis to highlight other sectors – the electricity generation sector, industry, transportation – would each of those studies merit publication in Nature Communications? If not, then what is the motivation for publishing this? The authors have more work to do, in my opinion, to justify why a focus on residential emissions should be published in a top journal, especially when they show that the contribution of residential emissions to the total air pollution is decreasing.

Response

The reason we focused on the residential sector is because the emission data of this sector have much higher uncertainty than those of other sources due to poor energy statistics for this sector compared with statistics for other sectors such as industry and power generation, which are better recorded by the government. Moreover, the rural residential energy use associated health impacts has been long overlooked. This is still the case even today. The recently published residential energy data, which demonstrate a rapid transition towards clean energy, provided us with a good opportunity to address this issue. This motivation, as well as the results that are very different from those previously reported, make this study unique and novel. In addition, the decreasing trend is due to energy transition, which shows important policy implications for future pollution control in China and other developing countries. These points were addressed in the rewritten introduction.

All comments and suggestions have been greatly appreciated. Please refer to the following responses to all specific comments.

General concerns:

Comment

1) The focus is on “rural Chinese households” or “rural residential”. Why is the focus only on residential emissions from rural regions? Why not include residential emissions from urban areas also? Or it seems that the focus is on solid fuel combustion, which is likely to be mostly in rural areas. But I thought that solid fuels are used for home heating and cooking in some urban areas, or if that is not common now, it would have been common in 1992 which is also modeled here. Why are urban emissions excluded here? And is it just solid fuels only, or does it include other residential fuels? More clarity is needed to better explain the focus, with some discussion to justify that choice.

Response

Most residential solid fuels, especially biomass fuels, are used in rural China, and the total consumption in rural areas was roughly estimated to be approximately 90%. Detailed rural residential energy consumption information derived from a recent national survey provides us with an excellent opportunity to evaluate the

environment, health, and climate impacts. Unfortunately, such detailed data for urban areas are not available at this time. For this reason, only the rural residential sector was evaluated in this study. In fact, we are working on data compilation for the urban residential sector, which will take time, mainly because of the difficulty in collecting high quality data on the energy use of rural-to-urban migrants.

The following sentence was added to the end of the Introduction: “Unfortunately, such detailed information is not yet available for urban areas to conduct such an evaluation.”.

The emission calculation was based on all residential energy types including LPG, biogas, and electricity. A sentence was added to the first paragraph to include this information: “The rural residential energy database covers detailed information of various energy types including coal, honeycomb briquette, straw, corncob, fuel wood, brushwood, charcoal, LPG, biogas, and electricity.”. In fact, emissions at power plants were also included based on electricity use.

Comment

2) The first two sentences of the abstract seem to contradict one another, and this contradiction shows up in many places in the paper. Residential emissions are a “missing piece of the puzzle” suggesting that they are not considered in air pollution control. But at the same time, there has been a remarkable energy transition that has reduced emissions substantially and has made the relative contribution of this sector decrease through time. It seems to me that language like “missing piece of the puzzle” is overselling the importance of this work.

Response

The rapid residential energy transition towards clean energy, which occurred spontaneously without intervention, did lead to reduced emissions. The remaining solid fuels used in this sector are still major emission sources. The first two sentences were rewritten as follows: “A recent energy survey showed that rural residential energy consumption is experiencing a rapid transition towards clean energy. Nevertheless, rural residential solid fuels remain as major energy types, especially those used for heating, and are an important emission source in China.”. The phrase “missing piece of the puzzle” was dropped accordingly.

The following sentences were included in the rewritten **Introduction**: “The results determined that solid fuel use for cooking and heating had been quickly replaced with liquefied petroleum gas (LPG), pipeline natural gas (PNG), biogas, and electricity, primarily due to rapid socioeconomic development. Similar rates have been reported by other surveys (Xie et al., 2014; Chen et al., 2016; Qiu et al., 2015). One direct result may be significant emission reductions of major air pollutants from this sector (Tao et al., 2018). Nevertheless, solid fuels remain as major energy types, especially for heating, leading to high emissions.”.

References

- Tao, S. et al. Quantifying the Rural Residential Energy Transition in China from 1992 to 2012 through a Representative National Survey. *Nature Energy*, 3, 567-573 (2018).
- Xie, Y. & Hu, J.W. An introduction to the China Family Panel Studies (CFPS). *Chinese Soc. Rev.* 47, 3-29 (2014).
- Chen, Y. L. et al. Transition of household cookfuels in China from 2010 to 2012. *Appl. Energ.* 184, 800–809 (2016).
- Qiu, H. G. et al. Renewable energy consumption in rural China: current situation and major driven factors. *J. Beijing Inst. Technol.* 17, 10–15 (2015).

Comment

3) When a conversion is made from solid fuels to LPG or electricity, emissions are reduced, but they do not go to zero. It is not clear to me whether these emissions are included in the analysis of changes in residential emissions, but I would guess that they are not. More clarity is needed on what is modeled, and if these increased emissions from LPG and electricity are neglected then some discussion of this point is needed.

Response

Emissions from all residential energy types including LPG and electricity, which was allocated to power plant sites, were included. The following sentences were added to the first paragraph of **Methodology**:

“Air pollutant emissions for the rural residential sector were mainly based on two recently updated databases. A new energy database was established and mainly based on a rural residential energy survey that covered detailed energy use activities, including staple food cooking, side dish preparation, water boiling, animal feed heating, and space heating for more than 34,400 households and daily biomass fuel consumption quantities for more than 1,600 households (Tao et al., 2018). The rural residential energy database covered detailed information on various energy types including coal, honeycomb briquette, straw, corncob, fuelwood, brushwood, charcoal, LPG, biogas, and electricity. The emission factor database was re-compiled based on carefully screened data from published studies conducted in China.”.

Comment

4) In addition to comparing results of the number of deaths with Lelieveld et al (ref. 7), the results should be compared with others in the literature – Silva et al (2016, EHP), Chafe (2014, EHP), Liang (2018, ACP). The authors found a relatively smaller contribution of residential than Lelieveld, but it is not clear to me that this is a fair comparison because of definitions of “residential” – whereas Lelieveld considers all residential emissions, the focus here seems to be on rural emissions and on solid fuel (or are other rural residential emissions included?).

Related to this, the smaller contribution compared with Lelieveld is interesting and suggests more analysis and discussion. The authors say that Lelieveld’s emissions (from commonly used global datasets) are too high because they don’t adequately reflect the energy transition which is captured here. Here the authors implicitly seem to think that their inventory is better than the global inventories, but they have not justified that conclusion.

Response

More comparisons were added, and the sentences from line 220 to 230 (“It was previously reported that 32% ... due to the nonlinearity of the dose-response curves”) were rewritten as follows and moved to the end of the paragraph. In the second half of the revised content below, the difference between our results and Lelieveld’s was further discussed, and we tried not to judge which one was more accurate in the revised version: “Sector emission associated premature deaths have been investigated previously (Butt et al., 2016; Chafe et al., 2014; Silva et al., 2016; Lelieveld et al., 2015; Liang et al, 2018). The differences among these studies are expected and can be explained by the differences in inventories, models, and methods adopted in health impact evaluation. Although Chafe et al. (2014) reported relatively small numbers of premature deaths in China (220,000, 170,000, and 130,000 in 1990, 2005 and 2010, respectively), because only cooking fuels were considered, a similar decreasing trend was demonstrated. Two other studies (Silva, et al., 2016, Liang et al., 2018) focused on the health impact of sector emissions on a global scale and reported in 2005 that a total of 207,000 deaths in China could be avoided if global residential emissions were zeroed, and that 26,000 deaths could be avoided if emissions from this sector were reduced by 20% in 2010. Lelieveld et al. (2015) reported that a total of 434,000 premature deaths, which was approximately 32% in terms of relative contribution, were attributable to PM_{2.5} and O₃ originating from residential emissions in 2010 in China. Considering that the contribution from residential sources to O₃ precursors was much smaller than that from other sectors (Li et al., 2017) and that residential emissions in China were predominantly from rural areas (Huang et al., 2014; Zhu et al., 2018), this value is higher than our estimate. This difference can be partially explained by the difference in emission inventories. Our inventory was characterized by a rapid increase in clean energy from 1992 to 2012 (Tao et al., 2018), a trend very different to that of previous data.”.

References

- Butt, E.W. et al. The impact of residential combustion emissions on atmospheric aerosol, human health, and climate. *Atmos. Chem. Phys.* 16, 873-905 (2016).
- Chafe, Z. et al. Household cooking with solid fuels contributes to ambient PM_{2.5} air pollution and the burden of disease. *Environ. Health Perspect.* 122, 1314-1320 (2014).
- Huang, Y., et al., Quantification of global primary emissions of PM_{2.5}, PM₁₀ and TSP from combustion and industrial process sources. *Environ. Sci. Technol.* 48, 13834-13843 (2014).
- Lelieveld, J., Evans, J.S., Fnais, M., Giannadaki, D. & Pozzer, A. The contribution of outdoor air pollution sources to premature mortality on a global scale. *Nature* 525, 367-371 (2015).
- Li, M., et al., MIX a mosaic Asian anthropogenic emission inventory under the international collaboration framework of the MICS-Asia and HTAP. *Atmos. Chem. Phys.* 17, 935-963 (2017)
- Silva, R. A., Adelman, Z., Fry, M., & West, J. J. The impact of individual anthropogenic emissions sectors on the global burden of human mortality due to ambient air pollution. *Environ. Health Perspect.* 124, 1776-1784 (2016)
- Tao, S. et al. Quantifying the Rural Residential Energy Transition in China from 1992 to 2012 through a Representative National Survey. *Nature Energy*, 3, 567-573 (2018).
- Liang, C. et al. HTAP2 multi-model estimates of premature human mortality due to intercontinental transport of air pollution and emission sectors. *Atmos. Chem. Phys.* 18, 10497-10520 (2018).

Comment

5) The atmospheric modeling does not seem to be new here, but is the results of references 28 and 29. Or if new modeling was done in this study (such as for the earlier years), that is not entirely clear. So this might reduce the novelty of this paper to the health and climate impacts.

Response

Although the modelling process followed Shen et al. (Ref. 28), the models were run for this study for all years using specifically developed emission scenarios, which were compiled based on a new rural residential energy database. The reference cited here serves to refer to the detailed modelling methodology and to support the model validation. This has been clarified by adding the phrase “in this study” to the

beginning of the first sentence of the **Atmospheric Chemical Transport Modelling** section in **Methodology**.

Comment

6) I don't know what the "exposure concentrations" mean. It says it is population-weighted, but does that mean conc x population in each grid cell? The units of Figure 3 bottom row show values between 10^{-5} and 10^{-10} , with units of $\mu\text{g m}^{-3}$, which would not seem to make sense with that definition. Should the units be "people $\mu\text{g m}^{-3}$ "?

Response

The term "exposure concentration" was replaced with "population weighted concentration" or "PWC" in short. The definition was added to line 66.

For PWC mapping (Fig. 3), the value of each grid cell equals the product of the model-calculated concentration and the population of an individual grid cell, which is then divided by the average population of all grid cells. This term is an intermediate variable for calculating the average PWC of the entire population with the unit $\mu\text{g/m}^3$. The spatial distribution reflects the exposure risk considering population density.

Fig. 3 Spatial distribution of annual mean model calculated $\text{PM}_{2.5}$ concentrations (A) and population-weighted $\text{PM}_{2.5}$ concentrations (B) attributed to rural residential emissions in 2012 in China. Source data are provided as a Source Data file.

Comment

7) The methods of assessing radiative forcing are entirely unclear to me. A radiative forcing is a difference between two states – it is not clear what the years in figure 5 are evaluated relative to (no residential emissions)? Does Figure 5 show annual global average forcing? In a single year from emissions changes, or in some future year representing steady state? Is the forcing accumulated over many years? I had thought that the authors might use the results of the WRF-Chem simulations to drive regional radiative forcing estimates, but that does not appear to be what is done here – rather a biogeochemical model and a "normalized marginal method" is used. I am not familiar with the marginal method, although it may be sufficient for this purpose. I don't understand what the biogeochemical model is used for – perhaps something with the carbon cycle. Critical here is what is assumed for CO_2 – whether the biomass is sustainably harvested or not – and that assumption is not at all clear. In line 256 it says that most fuel wood is carbon neutral – is that what is assumed in this study? In lines 260-262, it says that increase LPG and electricity consumption were compensated for by a decrease in fuel wood consumption – so do the results show both changes, and was this also done for the health calculations? Why are only these species evaluated and not, for example, the emissions of ozone precursors such as CO and NO_x ?

Response

Apologies for not clearly presenting the Methodology, which was revised to address each concern individually.

The forcing was the difference between the two states that include all emissions and that include all but residential emissions. The results were calculated for the four study years of 1992, 2002, 2007, and 2012 in China. The results are annual means.

The first sentence of the paragraph (lines 245-246) was revised as follows: "The annual mean radiative

forcing values attributable to rural residential emissions in China were calculated for the four study years as the differences between the radiative forcing derived based on all anthropogenic emissions and that based on all but rural residential emissions using the OSCAR model (see Methodology).” A sentence was added before “The derived climate forcing components include CO₂, BC, POA, sulfate, and nitrate” on line 323: “Again, the contributions from the rural residential and other sources were normalized using a proportional method.”

The forcing was calculated for individual years as annual means. The first sentence of the paragraph describing forcing calculation in Methodology (lines 318-325) was revised to “

The annual mean radiative forcing attributable to rural residential emissions was the difference between two scenarios of all anthropogenic emissions with and without rural residential sources. The global biogeochemical cycle model OSCAR v2.1 (Gasser et al., 2017; Cherubini et al., 2014) and the normalized marginal method was used (UNFCCC, 2002).”

The OSCAR model is a reduced-form model used widely in the climate change research community. Although it is a simple box-model without included seasonal variation, it is very efficient in terms of computation. As a parametric model, most of the parameters have been previously calibrated with complex models during a meta-modelling process. The last sentence of the section “**Health and Climate Impact Assessment**” in **Methodology** was replaced with “The OSCAR model is a reduced-form model used widely in the climate change research community (Li et al., 2016). Although this is a simple box-model without seasonal variation, it is very efficient in terms of computation. As a parametric model with most parameters previously calibrated against complex models during a meta-modelling process (Gasser et al., 2017), OSCAR fit our purpose well.”

For CO₂ emissions from biomass fuels, the following sentences were added to the **Methodology** (after “from 1992 to 2014” on line 289) to describe the calculation methods. “For CO₂ emission calculation, all crop residues were considered carbon-neutral, and for fuelwood, the non-renewable fractions for all individual provinces in China were sourced from the literature (Ru et al., 2015; Bailis et al, 2015).”

Both increased CO₂ emissions from LPG and electricity use and decreased CO₂ from biomass fuel use during the transition were calculated, and we found that the latter was larger than the former. Therefore, the net CO₂ emissions have decreased. The sentence was revised to: “Although the CO₂ emissions from LPG and electricity consumption increased, the decrease in CO₂ emissions from fuelwood use was faster, leading to a net reduction in CO₂ emissions from this sector from 1992 to 2012.”

Similarly, changes in ambient PM_{2.5} concentrations due to decreases or increases of both primary PM_{2.5} and secondary PM_{2.5} precursors associated with the changes in various fuel types during the transition were quantified individually to obtain the net change for health impact assessment. Therefore, the net impacts on premature death could be derived (see lines 299-309 in **Methodology**).

The species used for climate forcing calculation were chosen based on OSCAR. The health impact was focused on PM_{2.5} only because emissions of major O₃ precursors (NO_x and VOCs) are predominantly from non-residential sources in general and emission data of VOCs from the residential sector are scarce. On the other hand, CO impact occurs mainly in the indoor environment. The following sentence was added after the third sentence of “**Health and climate co-benefits of the rural residential energy transition**” to clarify that only the PM_{2.5} associated health impact is assessed in this study: “The health impacts of other pollutants such as O₃ and CO were not evaluated in this study due to the relatively low impacts originating from this sector and a shortage of emission data.”

References

- Tao, S., et al., Quantifying the rural residential energy transition in China from 1992 to 2012 through a representative national survey, *Nature Energy*, 3, 567-573 (2018).
- Bailis, R., Drigo, R., Ghilardi, A., & Masera, O. The carbon footprint of traditional woodfuels. *Nature Climate Change*, 5, 266-272 (2015).
- Li, B., et al., The contribution of China’s emissions to global climate forcing. *Nature*, 531, 357-361 (2016).
- Ru, M., et al. Direct energy consumption associated emissions by rural-to-urban migrants in Beijing. *Environ. Sci. Technol.* 49, 13708-13715 (2015).
- Gasser, T., Ciais, P., Boucher, O., Quilcaille, Y., Tortora, M., Bopp, L., & Hauglustaine, D. The compact Earth system model OSCAR v2.2: description and first results. *Geosci. Model Dev.* 10, 271-319 (2017).

Comment

8) The model evaluation given (Figs. S5 and S6) is pretty bare bones with no statistical summaries. Could the authors model 2015 for better comparison with the observations that were available in 2015 than in 2012?

Response

For both Figures S5 and S6, the normalized mean bias (NMB) and normalized mean error (NME) were calculated and added accordingly to the figures in the revised version. The following sentences were added to line 299: “Normalized mean bias (NMB) and normalized mean error (NME) were calculated for all validations. The model calculations generally agree with the observations in all validation cases.”. Apologies, we do not have the emission inventories needed for a 2015 simulation.

Comment

9) The study ends without conclusions – the last paragraph is almost entirely about the fact that household exposure is not addressed here.

Response

A paragraph was added to the end of the **Results and Discussion** as a concluding remark (a conclusion section is not allowed by the journal):

“In summary, a rapid transition of rural residential energy away from solid fuels from 1992 to 2012 resulted in significant reductions in the contribution of this sector to ambient PM2.5 concentrations, subsequently reducing both adverse health impacts and climate forcing. Despite these changes, solid fuels are still used extensively in rural China for heating, contributing significantly to air pollution and population exposure.”.

Comment

10) The paper seems long to me for a Nature journal. There is a lot of long discussion of results that I don't think are very hard to understand (the difference between absolute and relative results), repetition of results (different in winter), and statements of policy relevance scattered throughout, so I have the sense that the whole paper could be conveyed in much fewer words if needed. But if the length is acceptable with the journal, then I have no problem.

Response

Some sentences were deleted accordingly (Lines 105-108, 114-115, 117-121, 151-156, 164-166, and 196-202).

More specific comments:

Comment

l. 10 – I think that the deaths should be “per year”. This needs to be clear throughout the paper.

Response

Added accordingly.

Comment

l. 11 – a 20 year period – when?

Response

The text was replaced with “from 1992 to 2012”, accordingly.

Comment

l. 40-46 – This is good, but it has not been shown that the China-based inventory is more accurate than the international ones. I think it is likely better, but it has not been demonstrated to be more accurate, and no further argument is given that it is better.

Response

The slow change rate (0.5%/year) reported by the IEA differs greatly from the experiences of the Chinese majority. This rapid transition was also confirmed by the results of three local studies, which have been added to the references section in the revised version. A sentence was added after “... rate of approximately 5%.”: Similar rates have been reported by other surveys (Chen et al., 2016; Qiu et al., 2015; Xie and Hu, 2014).”.

References

Xie, Y. & Hu, J.W. An introduction to the China Family Panel Studies (CFPS). *Chinese Soc. Rev.* 47, 3-29 (2014).
Chen, Y. L. et al. Transition of household cookfuels in China from 2010 to 2012. *Appl. Energ.* 184, 800–809 (2016).
Qiu, H. G. et al. Renewable energy consumption in rural China: current situation and major driven factors. *J. Beijing Inst. Technol.* 17, 10–15 (2015).

Comment

l. 66 – The authors should more carefully define “ambient” and “population-weighted”. I don't think ambient is usually used in this way.

Response

The phrase “ambient air” was replaced with “the model calculated”

Comment

l. 124 – The discussion around Fig. 2 is based on this graph of only one city in one month. It leaves the question of whether this is representative of other locations and months. The authors have the data to analyze much more than this.

Response

The high daily variation is common in most areas of China. Hebei is a province rather than a city and is one of the most polluted regions in China. The phrase “as an example” was replaced with “...” and the following sentence was also added: “This finding was common for most grid cells, and the calculated coefficients of variation for all individual grid cells in mainland China in 2012 were as high as $70\pm 35\%$.”

Comment

l. 151-153 – This suggests that coal and biomass were used very recently in urban areas, bringing into question why this isn’t modeled.

Response

The campaign is primarily for rural areas. The residential emissions from urban areas were only approximately 10% of those from the entire residential sector. Additionally, detailed urban residential energy consumption data are not available, primarily due to millions of rural-to-urban migrants who use very different and diverse energy types compared with permanent urban residents. These sentences were deleted in the revision given the limited space permitted by the journal.

Comment

l. 156 – I don’t understand the text.

Response

This phrase was deleted in the revision based on another comment.

Comment

l. 208 – This is not how the word co-benefits is usually used. The authors should define what is a co-benefit of what. And this section doesn’t really evaluate benefits – rather it evaluates trends through time.

Response

Both reductions in health impacts and climate forcing were driven by the rural residential energy transition towards clean energy, which we refer to as health and climate co-benefits of the residential energy transition. Please correct us if this is incorrect.

Comment

l. 292 – I think 2009 should be 2007.

Response

Apologies for the mistake, which was corrected accordingly.

Comment

l. 297-299 – Rather than just saying that it was evaluated against measurements, it would be better to summarize the main findings of that comparison.

Response

The sentence “The model performance was previously evaluated.” was revised to “The model performance was previously evaluated with reasonable success.”.

The following sentence was added after “... for 367 cities in 2015 (Fig. S6).”: “Normalized mean bias (NMB) and normalized mean error (NME) were calculated for all validations. The model calculations generally agree well with the observations in all validation cases.”.

In addition, another set of validation on PM_{2.5} components was added, and a sentence was added to line 299: “The model was also validated for major PM_{2.5} components, including black carbon, organic carbon, and SNA (sum of sulfate, nitrate, and ammonium). The results are shown in Fig. S7.”.

Fig. S7 Comparison of the calculated concentrations of major $PM_{2.5}$ components including black carbon (A), organic carbon (B) and SNA (sum of sulfate, nitrate, and ammonium) (C) with those observed. The dashed and dotted lines represent the error ranges of two and five times, respectively. Normalized mean bias (NMB) and normalized mean error (NME) are also shown. Source data are provided as a Source Data file.

Comment

Fig. 2 – When is this shown? What is the x-axis?

Response

This was January (from Jan. 1st to 31st) as indicated on line 125.

Comment

Fig. 3 – A logarithmic scale may be appropriate here, but I find $10^{1.6}$ to be not very useful on the color scale – better to show the actual numbers. I don’t understand what is plotted in the bottom row or its units.

Response

Fig. 3 has been redrawn in the revision according to reviewer comments, and the legend numbers are the actual numbers of ambient and population-weighted concentrations. The bottom panels are part of the base map.

Fig. 3 Spatial distribution of annual mean model calculated $PM_{2.5}$ concentrations (A) and population-weighted $PM_{2.5}$ concentrations (B) attributed to rural residential emissions in 2012 in China. Source data are provided as a Source Data file.

Comment

Fig. 4 – in panel a, does the blue at the top signify anything? In panel b, what is “the attributed factor” (is this the “attributable fraction”)? Why does 1992 look like lower concentrations than 2012, when other graphs show decreases through time?

Response

In panel A, the blue was background, which was removed. In panel B, the phrase was corrected to “attributable fraction”. The concentration in panel B is from all sources, which was higher in 2012 than in 1992 due to increased emissions from all other sectors but the residential sector (lines 231-234).

Additionally, considering comments from the other reviewers, the figure was revised to:

Fig. 4 Model-calculated population risk associated with exposure to PM_{2.5} originating from rural residential emissions from 1992 to 2012 (**A**); and **B**) the nonlinear dose-response relationship and the cumulative frequency distributions of the total PWC from all sources in 1992 and 2012. The results are shown for chronic obstructive pulmonary disease (COPD), cerebrovascular disease (stroke), ischaemic heart disease (IHD), lung cancer (LC), and acute lower respiratory infections (ALRIs). Source data are provided as a Source Data file. Source data are provided as a Source Data file.

Reviewers' comments:

Reviewer #1 (Remarks to the Author):

The authors have done a good job in addressing my concerns. I am impressed by the recalculations of the health impacts using the new framework.

Minor suggestion here:

sulfate, nitrate and ammonium has different levels of climate impacts, I would suggest doing comparisons for each instead of using the total.

Reviewer #3 (Remarks to the Author):

Air Pollution from Rural Chinese Households under Fast Residential Energy Transition

Summary

Thanks to the authors for revising the manuscript and for confirming and fixing numerous errors throughout.

I still have a significant concern with this manuscript in its current form. The authors state in Lines 321 – 322 that the meteorological model was not evaluated for this study, and refer to previous citations of WRF application in China. For e.g. the Wang et al, AE 2016 was for WRF application at 36x36 and 12x12 km resolutions in January 2013 alone. Whereas the Zhao et al, 2012 paper is for a limited region over East China and neighboring countries in forecast mode for NWP at different resolutions. Given the role of meteorology in air quality model predictions, this is inadequate. Why do the authors believe that WRF evaluation performed elsewhere for other studies suffices for this study? To gain confidence in the AQ/health/climate results presented here, one should first gain confidence in the meteorological fields that were used, by evaluating year-specific model predictions for the 4 years modeled in this study with corresponding observations available in the entire modeling domain. Specifically, was there any potential bias in WRF meteorology in rural vs urban areas that may feed into the findings here?

Few more comments:

- a) Figure 1: What explains high absolute PWC for urban but low % contribution in right panel, in all 4 years? Also, is the % estimate for PWC or just ambient concentrations?
- b) Figure 3: Please remove insets. They are poor quality, and add no value
- c) Figure 4 Caption: Please remove duplicate entry of "Source data are provided as a Source Data file."
- d) Figure 5: In left panel y-axis, please change "Forcing" to "Radiative Forcing"
- e) Lines 80-81: Shouldn't PWC have units of "people-ug/m3"? Also, instead of "model calculated concentration", suggest "air quality concentration"
- f) Lines 283 – 284: "This constraint is not limited to this study but is also shared by nearly all past studies on the health impacts of air pollution." Didn't the recent GBD studies address this? Also I find that the comment that this is due to "lack of quantitative data on fugitive emissions" is too narrow. Aren't there other gaps? Please clarify
- g) Line 286: "... replacing the remaining solid fuels ..." Please clarify with what?
- h) Lines 297 – 298: Besides PM25 and PM10 does WRF/Chem treat TSP independent of these, and output them?
- i) Line 322-323: "The model performance was previously evaluated with reasonable success". This implies that the activity of evaluation was successful, but that doesn't convey much.
- j) Lines 324-325: "The model calculations generally agree with the observations in all validation cases". This sentence is not providing any useful information. Suggest that the authors provide additional insights into the findings from the model evaluation. Specifically, please mention areas where the model is strong, and where it is weak, based upon all evaluation that was done for this

study.

k) References: Please use consistent style for all references. Some of them use "et al" while others have all co-authors listed.

Response to comments

Title: Air Pollution from Rural Chinese Households under Fast Residential Energy Transition

Response to editor's comments

Comment

Your manuscript entitled "Impacts of Air Pollutants from Rural Chinese Households under the Rapid Residential Energy Transition" has now been seen by the referees. You will see from their comments below that the referees acknowledge the improvement of your revised work, but between them they also raise a number of concerns, which must prevent us from offering to publish the paper in its present form. We therefore invite you to send a revised manuscript that addresses all of the referees' concerns, as well as point-by-point responses to the referees' comments that you are happy for us to return to the referees along with a revised manuscript.

Response

We greatly appreciate the comments provided by the three reviewers that have helped us further clarify and improve the manuscript. The manuscript has been revised accordingly and point-by-point responses have been submitted along with the revised manuscript.

Comment

The referees' reports seem to be quite clear. Naturally, we will need you to address all of the points raised. Specifically, for publication in Nature Communications to be appropriate, we will need you to provide more compelling validation and justifications to the meteorological model used in your study (Reviewer #3). At the same time, we will need you to make comparisons regarding sulfate, nitrate and ammonium impacts on climate (Reviewer #1).

Response

All these comments including those provided by the fourth reviewer, as well as other detailed comments, have been addressed in the revised version. Please refer to the detailed point-by-point responses to specific comments and the revised manuscript.

Response to reviewer's comments

All line numbers referred to in these responses are from the previous version (NCOMMS-18-25418A).

Reviewer-1

Comment

The authors have done a good job in addressing my concerns. I am impressed by the recalculations of the health impacts using the new framework.

Minor suggestion here: sulfate, nitrate and ammonium has different levels of climate impacts, I would suggest doing comparisons for each instead of using the total.

Response

In climate forcing estimation using the reduced-form model, components considered in the study include CO₂, BC, POA, sulphate, and nitrate (lines 355-356). In response to this comment, we compared the modelled concentrations with the observations for sulfate, nitrate and ammonium separately in the revised version. The sentence on line 323 was revised as: "The model was also validated for major PM_{2.5} components, including black carbon, organic carbon, sulfate, nitrate, and ammonium (Fig. S8)." and Figure S8 (Figure S7 in the previous version) was replaced with the following:

Fig. S8 Comparison of the modelled concentrations of major PM_{2.5} components including black carbon (A), organic carbon (B), sulfate (C), nitrate (D), and ammonium (E) with those observed. The dashed and dotted lines represent the error ranges of two and five times, respectively. Sample size (N), normalized mean bias (NMB) and normalized mean error (NME) are also shown. Source data are provided as a Source Data file.

Comment

Summary: Thanks to the authors for revising the manuscript and for confirming and fixing numerous errors throughout. I still have a significant concern with this manuscript in its current form. The authors state in Lines 321 – 322 that the meteorological model was not evaluated for this study and refer to previous citations of WRF application in China. For e.g. the Wang et al, AE 2016 was for WRF application at 36x36 and 12x12 km resolutions in January 2013 alone. Whereas the Zhao et al, 2012 paper is for a limited region over East China and neighboring countries in forecast mode for NWP at different resolutions. Given the role of meteorology in air quality model predictions, this is inadequate. Why do the authors believe that WRF evaluation performed elsewhere for other studies suffices for this study? To gain confidence in the AQ/health/climate results presented here, one should first gain confidence in the meteorological fields that were used, by evaluating year-specific model predictions for the 4 years modeled in this study with corresponding observations available in the entire modeling domain. Specifically, was there any potential bias in WRF meteorology in rural vs urban areas that may feed into the findings here?

Response

We understand the reviewer's concern about the confidence for the meteorological input and we fully agree with the reviewer that the meteorological data driving atmospheric chemistry and transport is essential for ensuring accurate simulation of chemistry. We hope the following arguments could address the reviewer's concern. First, WRF is an operational weather forecasting model that has been extensively evaluated and verified by numerous studies. For a forecasting model, its operational application would not be approved unless sufficient accuracy has been demonstrated and reached. Second, WRF does not input directly measured raw meteorological data as the initial conditions but uses "objectively analyzed meteorological data". Data objective analytical system is a very complex data processing system taking into consideration of measured raw meteorological data, satellite remote sensing data, other instrument sampled meteorological data, and data assimilation. During the model performance, available observed meteorological data are implemented into the objectively analysed data system (at 00, 06, 12, and 18 local time) and then input into WRF after data processing to re-initialize the model integration and to correct potential forecasting errors. With this mechanism, WRF forecasted meteorology is continuously updated by objectively analyzed raw meteorological data. Due to such frequent verification (every six hours) based on true observations, one would expect no significant bias between the simulated and real meteorology for the entire performance for a "past" simulation with observational data available. In our case, the WRF-Chem model was not used to forecast future meteorology and air quality, but to study the "past" air quality driven largely by known meteorology in which the input (measured) and output (modeled) meteorology is, to a large extent, consistent. Even these hourly output and input data are averaged annually, they are still consistent. In fact, this is the main reason that the evaluation of an operational forecasting model is often conducted by examining its capability to forecast a weather "event", e.g., precipitation, cold break, instead of the measured meteorological data, or for its accuracy in predicting future meteorology (e.g., air temperature).

According to the comments, the input meteorology was evaluated against field observations from the China National Meteorological Information Centre. The sentence on lines 316-317 was revised as: "The WRF meteorological inputs were evaluated against observations from China Earth International Exchange Stations. The normalized mean bias (NMB) and normalized mean error (NME) were

calculated for all validations. The NMB for surface pressure, temperature, relative humidity and wind speed were -3%, -7% ~ -10%, -2% ~ -9% and 32% ~ 43%, respectively (**Table S2**).”.

Table S2. Statistics for evaluating the daily mean surface pressure, temperature, relative humidity and average wind speed for the four study years between the WRF meteorological inputs and observations. Source of observed data: National Meteorological Information Center, China Earth International Exchange Station Climate Data-Daily Value Data Set (V3.0), <http://data.cma.cn/>

	Pressure			Temperature			Relative Humidity			Wind Speed		
	N	NMB	NME	N	NMB	NME	N	NMB	NME	N	NMB	NME
1992	60389	-3%	3%	60390	-9%	21%	60390	-3%	16%	57168	39%	56%
2002	60225	-3%	3%	60223	-11%	20%	60224	-2%	15%	57259	43%	59%
2007	60590	-3%	3%	60590	-7%	17%	60590	-9%	18%	60024	32%	48%
2012	60747	-3%	3%	60747	-9%	19%	60738	-7%	16%	60720	32%	48%

Comment

a) Figure 1: What explains high absolute PWC for urban but low % contribution in right panel, in all 4 years? Also, is the % estimate for PWC or just ambient concentrations?

Response

The relatively high absolute PWC for urban areas is due to relatively high population densities because PWC is population weighed. The relatively low contribution of the residential sector to the overall PWC from all sources is due to relatively high emissions from other sectors such as industry and transportation in urban areas.

To explain this concept, the following phrase was added on line 95: “..., due to strong emissions from other sectors in urban areas”.

The percentages estimated in Fig. 1B were for PWC instead of ambient air concentrations. To clarify this, the y-axis label in Fig. 1B was revised as “Relative contribution for PWC”, and the figure caption was revised as: “... to the overall population-weighted PM_{2.5} concentration (PWC) in mainland China...”.

Comment

b) Figure 3: Please remove insets. They are poor quality, and add no value

Response

The inset, which is part of the background map, was replaced with a high-resolution version.

Comment

c) Figure 4 Caption: Please remove duplicate entry of “Source data are provided as a Source Data file.”

Response

Removed accordingly.

Comment

d) Figure 5: In left panel y-axis, please change “Forcing” to “Radiative Forcing”

Response

Revised accordingly.

Comment

e) Lines 80-81: Shouldn't PWC have units of "people-ug/m3"? Also, instead of "model calculated concentration", suggest "air quality concentration"

Response

To calculate the PWC of a region with a number of sub-regions (grids in this study), the concentration-population products of the sub-regions were summed and then divided by the total population of the region. Therefore, the PWC unit is still $\mu\text{g}/\text{m}^3$. The following sentence was added on line 337 in the Method section to clarify this point: "The PWC of a region with more than one grid cell was calculated as $\Sigma(P_i \cdot C_i)/P_t$, where P_i and C_i are population and air $\text{PM}_{2.5}$ concentration at the i th grid, and P_t is the total population in the domain of interest.". The phrase "model-calculated concentration" on lines 76, 329 and 367, and the phrase "the model-calculated $\text{PM}_{2.5}$ concentration" on lines 199, 336, and 341 were replaced by "air quality concentration".

Comment

f) Lines 283 – 284: "This constraint is not limited to this study but is also shared by nearly all past studies on the health impacts of air pollution." Didn't the recent GBD studies address this? Also I find that the comment that this is due to "lack of quantitative data on fugitive emissions" is too narrow. Aren't there other gaps? Please clarify

Response

As stated in the latest GBD study, no substantial change has been made in modelling household air pollution (Stanaway et al., 2018). It has been modelled using a three-step strategy, including covariates such as maternal education, proportion of population living in urban areas. This indirect method introduced higher uncertainties compared with ambient air quality modelling, especially given limited field observations and large variations of indoor air concentrations among individual households. The sentences on lines 276-279 were revised as "..., household exposure is subject to limitations and constraints at this time. These constraints and limitations are borne by almost all existing literature on the health impacts of indoor air pollution, including the latest GBD study (Stanaway et al., 2018). There is an urgent need to develop a reliable method to characterize ...".

The phrase on line 275 was rephrased as "... due to the lack of quantitative data on parameters including fugitive emission, air exchange rate, indoor circulation, etc."

References

Cohen, A.J. et al., Estimates and 25-year trends of the global burden of disease attributable to ambient air pollution: an analysis of data from the Global Burden of Diseases Study 2015. *Lancet* 389, 1907-1918 (2017).
Stanaway, J. D., et al. Global, regional, and national comparative risk assessment of 84 behavioural, environmental and occupational, and metabolic risks or clusters of risks for 195 countries and territories, 1990–2017: a systematic analysis for the Global Burden of Disease Study 2017." *The Lancet* 392, 1923-1994 (2018).

Comment

g) Line 286: "... replacing the remaining solid fuels ..." Please clarify with what?

Response

A phrase "with affordable cleaner fuels or electricity" was added after "... replacing the remaining solid fuels".

Comment

h) Lines 297 – 298: Besides PM_{25} and PM_{10} does WRF/Chem treat TSP independent of these, and output them?

Response

The WRF/Chem model has different chemical components with separate tracers (sulfate, nitrate, black carbon, organic matter, etc.) to simulate different size bins of particulate matter, and the data from different tracers are summed to calculate corresponding particulate matter fractions (i.e. PM_{2.5}, PM₁₀, and TSP). The MADE/SORGAM aerosol scheme used in this study does not have dust (TSP) output.

Comment

i) Line 322-323: “The model performance was previously evaluated with reasonable success”. This implies that the activity of evaluation was successful, but that doesn’t convey much.

Response

The sentence was deleted, and the sentence on lines 317-319 was revised as: “The model performance was validated against the observations reported by ...”.

Comment

j) Lines 324-325: “The model calculations generally agree with the observations in all validation cases”. This sentence is not providing any useful information. Suggest that the authors provide additional insights into the findings from the model evaluation. Specifically, please mention areas where the model is strong, and where it is weak, based upon all evaluation that was done for this study.

Response

Although discrepancies often occur randomly in space, there are still some recognizable patterns. A detailed description of the pattern was added to the discussion. The sentences on line 324 were revised as: “Although the simulated results positively biased for total PM_{2.5} (NMB=25% and 30% for Beijing and Shanghai, respectively), black carbon (NMB=67%), nitrate (NMB=49%), and ammonium (NMB=38%), and negatively biased for particulate sulfate (NMB=-12%), the model predicted temporal and spatial variations of ambient PM_{2.5} are generally acceptable. The discrepancy is very likely from uncertainties in model inputs, missing mechanisms in the model (Tuccella et al., 2012; Wang et al., 2016), as well as high uncertainties in limited particle composition observational data with different sampling and analytical methods. The bias showed a spatial pattern with positive and negative values in eastern and western China, respectively, which are illustrated in **Fig. S9**. The overestimation in eastern China is attributed to the fact that the model was run for 2012, whereas observational data were collected in 2015 and a significant decrease has occurred since 2014. On the other hand, the influence of dust in the west is likely one important reason for the underestimation in this region.”.

Meanwhile, the detailed composition validation was revised to compare individual components separately and the results are provided in **Fig. S8**.

References

- Tuccella, P.; Curci, G.; Visconti, G.; Bessagnet, B.; Menut, L.; Park, R. Modelling of gas and aerosol with WRF/Chem over Europe: evaluation and sensitivity study. *J. Geophys. Res.* 117, D03303 (2012)
- Wang, L.; Zhang, Y.; Wang, K.; Zheng, B.; Zhang, Q.; and Wei, W. Application of Weather Research and Forecasting Model with Chemistry (WRF/Chem) over northern China: sensitivity study, comparative evaluation, and policy implication. *Atmos. Environ.* 124, 337-350 (2016).
- Zheng, B. et al., Heterogeneous chemistry: a mechanism missing in current models to explain secondary inorganic aerosol formation during the January 2013 haze episode in North China. *Atmos. Chem. Phys.* 15, 2031-2049 (2015).

Fig. S8 Comparison of the modelled concentrations of major PM_{2.5} components including black carbon (A), organic carbon (B), sulfate (C), nitrate (D), and ammonium (E) with those observed. The dashed and dotted lines represent the error ranges of two and five times, respectively. Sample size (N), normalized mean bias (NMB) and normalized mean error (NME) are also shown. Source data are provided as a Source Data file.

Fig. S9 Distribution of the relative difference in the model calculated PM_{2.5} concentrations in 2012 in comparison with the observation in 2015. Source data are provided as a Source Data file.

Comment

k) References: Please use consistent style for all references. Some of them use “et al” while others have all co-authors listed.

Response

The reference style follows the journal guideline. All authors are included in the reference lists unless there are six or more, in which case only the first author is given.

Reviewer-4

Comment

This manuscript has improved from the first submitted version, through revisions in response to comments from reviewers. I remain concerned that the novelty of this paper is not made sufficiently clear, and that while in some cases the authors responded in a response to reviewers' document, no corresponding changes (or insufficient changes) were made in the main paper itself.

To me, the Introduction does not succeed in communicating the value or novelty of the paper. The Introduction cites a large number of related studies that do similar things (though perhaps for different sectors or countries), leaving the impression that this paper is just another one in a long line of similar papers. Starting from the second sentence of the paper (l. 25-44), the writing follows the "one sentence explanation of each study" format, which gives a good understanding of what came before this paper, but really fails to tie those papers together coherently and show clearly how this paper differs. Starting in l. 45, the attention turns to motivating "rural residential emissions", and describe new data that is now available to quantify this. This seems like a good basis to motivate this study, but the novelty or importance of this approach is not emphasized. To me, by failing to motivate why this paper is novel, I don't think this study achieves the standard of a high-profile journal like Nature Communications.

Response

We apologize for not making this point clear and not revising the manuscript sufficiently. Here further efforts were made following the reviewer's comments. The key points to demonstrate novelty and significance are:

- 1) The residential sector is one of the most important emission sources of major air pollutants in China, contributing significantly to air pollution and human health. The main reason is very high emission factors of air pollutants from residential stoves due to low burning efficiency and no end treatment;
- 2) Unfortunately, this sector has been overlooked for its contribution to air pollution in China by policy makers, the public, and even scientists. There are at least two reasons: 1) the energy consumption in this sector accounts for a relatively small fraction of the total, and 2) data on rural residential energy consumption and associated emissions have been poorly recorded and often assumed, in comparison with other sectors such as industry, energy production, and transportation.
- 3) Therefore, the residential sector has become the "shortest wooden bar" in the overall evaluation of emissions, air pollution, and health impacts, contributing significantly not only to air pollution, but also to the overall uncertainty in air pollution evaluation.
- 4) Recently, a nationwide rural residential energy survey was conducted in China, which, for the first time ever, provided rich and first-hand data. One major finding is that the total quantities of solid fuels used in this sector as well as temporal changes have been misestimated to a large extent in the past. This misestimation is particularly true for biomass fuels, which contribute approximately half of the total solid fuel use in rural households and are completely "forgotten" in all mitigation plans.
- 5) The new data provide a unique opportunity to improve our understanding of contributions from the rural residential sector to air pollution and consequent health and climate impacts in China. In fact, the relatively high uncertainty associated with the residential sector is a worldwide phenomenon.

In the revised version, the **Introduction** has been rewritten to convey these key messages.

Introduction

Air pollution is one of the most concerning environmental issues in China, causing more than one million premature deaths each year.¹⁻² Research has documented that air pollutants are primarily from power generation, industry, transportation, agriculture, and residential activity.³⁻⁴ Although residential fuel consumption contributes a very small fraction of the total energy use in China, it contributes significantly to pollutant emissions and consequently adverse health and climate impacts.^{2, 5-6} This phenomenon is because the emission factors (EFs, quantities of air pollutants emitted per unit of fuel consumed) for extensively used solid fuels in this sector are very high. According to the latest estimation, 27% and 51% of primary PM_{2.5} (particulate matter with an aerodynamic diameter less than 2.5 μm) and BC (black carbon) were emitted from the residential sector in mainland China in 2014 and nearly 80% was from rural areas.⁷

Unfortunately, without enough first-hand data, the important role that residential energy transition can play in mitigating health burdens has hardly been justified. The residential sector has long been overlooked for its contribution to air pollution in China,⁸⁻⁹ likely because the relatively low contribution of the residential sector to total energy use was known, whereas very high EFs are often not aware of. Another obstacle for better understanding is the fact that residential energy consumption and emissions were poorly recorded in comparison with other sectors,¹⁰⁻¹¹ leading to high uncertainty in emission estimation. Consequently, the residential sector is the “shortest wooden bar” in the overall evaluation of emissions and air pollution.

Recently, a thorough nationwide survey was conducted to collect first-hand data on rural residential energy use from 1992 to 2012 in rural China.¹¹ A rapid transition of rural residential energy from solid fuels towards cleaner energy was revealed. The study also found that the quantities of biomass fuel use and the energy mix transition have been misestimated to a large extent.¹¹ The new data from this survey provide us with a unique opportunity to improve our estimation of residential contributions to air pollution and to evaluate the impacts of the residential energy transition on health and climate forcing. In fact, the co-impacts of air pollution and co-benefits of emission reduction on health- and climate-relevant air pollutants are often expected and recognized in literature.^{5, 12-16} For example, studies quantified premature deaths avoided and the influence on radiative forcing induced by using low sulfur jet and ship fuels,¹³⁻¹⁴ and co-benefits from increasing household insulation in U.S. households have been previously reported.¹⁵

In this study, we quantitatively distinguish emissions from rural residential sources and all other sectors based on the newly compiled emission inventories of major air pollutants, model the contributions of the rural residential sector to ambient PM_{2.5}, and evaluate the co-impacts of the residential energy transition on air pollution-associated health and climate radiative forcing (see **Methodology**). The reason for focusing on rural areas is because the majority of residential emissions in China are from rural areas where the detailed residential energy survey was targeted.

References

1. Cohen, A.J. et al., Estimates and 25-year trends of the global burden of disease attributable to ambient air pollution: an analysis of data from the Global Burden of Diseases Study 2015. *Lancet* 389, 1907-1918 (2017).
2. Lelieveld, J., Evans, J.S., Fnais, M., Giannadaki, D. & Pozzer, A. The contribution of outdoor air pollution sources to premature mortality on a global scale. *Nature* 525, 367-371 (2015).
3. Huang, Y. et al. Quantification of global primary emissions of PM_{2.5}, PM₁₀, and TSP from combustion and industrial process sources. *Environ. Sci. Technol.* 48, 13834-13843 (2014).
4. Wang, R. et al. High-resolution mapping of combustion processes and implications for CO₂ emissions. *Atmos. Chem. Phys.* 13, 5189-5203 (2013).
5. Butt, E. W. et al. The impact of residential combustion emissions on atmospheric aerosol, human health, and climate. *Atmos. Chem. Phys.* 16, 873 (2016).
6. Chafe, Z. et al. Household cooking with solid fuels contributes to ambient PM_{2.5} air pollution and the burden of disease. *Environ. Health Perspect.* 122, 1314-1320 (2014).
7. Zhu, X., et al. Stacked use and transition trends of rural household energy in mainland China. *Environ. Sci. Technol.* 53, 521-529 (2019).
8. Liu, J. et al. Air pollutant emissions from Chinese households: a major and underappreciated ambient pollution source. *Proc. Natl. Acad. Sci. USA* 113, 7756-7761 (2016).
9. Sheng, G. & Gao, J. Ministry of Ecology and Environment: Three main factors contributing to severe air pollution. *Xinhua Daily Telegraph*. pp. A8. (Mar. 11, 2019)
10. Chen, Y. L. et al. Transition of household cookfuels in China from 2010 to 2012. *Appl. Energ.* 184, 800-809 (2016).
11. Tao, S. et al. Quantifying the Rural Residential Energy Transition in China from 1992 to 2012 through a Representative National Survey. *Nature Energy*, 3, 567-573 (2018).
12. Conibear, L. et al. Residential energy use emissions dominate health impacts from exposure to ambient particulate matter in India. *Nature Commun.* 9, 167 (2018).
13. Barrett, S. R. et al. Public health, climate, and economic impacts of desulfurizing jet fuel. *Environ. Sci. Technol.* 46, 4275 (2012).
14. Sofiev, M. et al. Cleaner fuels for ships provide public health benefits with climate tradeoffs. *Nature Commun.* 9, 406 (2018).
15. Levy, J. I. et al. Carbon reductions and health co-benefits from US residential energy efficiency measures. *Environ. Res. Letters* 11, 034017 (2016).
16. West, J. et al. Co-benefits of global greenhouse gas mitigation for future air quality and human health. *Nat. Clim. Chang.* 3, 885-889 (2013).

Comment

1) The authors now clarify what fuels are counted in rural residential emissions, but I'm not aware that they have defined what areas in China are rural, nor do I really see an explanation for why they choose to focus on rural residential except for the fact that a new study makes these data available.

Response

To clarify the method of distinguishing rural and urban areas in space, the sentence on line 333 in the **Method** was revised as: "Rural and urban areas were spatially distinguished by using a dynamic urban mask developed Shen *et al.*, (2017). In brief, the urban mask was developed by extracting built-up areas (urban) using remote sensing data, nighttime light data, and population census data, based on county-specific thresholds."

In China, rural areas are more important than urban areas in terms of contributions to residential emissions and uncertainty, because not only do 80% of residential emissions occur in rural areas (Zhu et al., 2019), but the uncertainty in residential emissions is also largely associated with biomass fuels, which are almost solely used in rural areas. Additionally, for this reason, the survey focused on rural China and equivalent detailed data are not available for urban areas at this time.

The last sentence of the Introduction was revised as: "The reason for focusing on rural area is because the majority of residential emissions in China are from rural areas where the detailed residential energy survey was targeted."

References

Shen, H.Z. et al. Urbanization-induced population migration has reduced ambient PM_{2.5} concentrations in China. *Sci. Adv.* 3, e1700300 (2017).

Zhu, X., et al. Stacked use and transition trends of rural household energy in mainland China. *Environ. Sci. Technol.* 53, 521-529 (2019).

Comment

2) For the 52% decline in mortality in the abstract, it seems that this is in spite of population increase over this time, but baseline mortality rates likely also changed. It would be interesting to see the contributions of PM from rural residential, population, and baseline mortality to this overall change.

Response

The decline presented was the net change in mortality from 1992 to 2012, reflecting the changing of population, exposure and baseline mortality rates. In the revised version, the relative contributions of population growth, population ageing, baseline mortality rates, and exposure were analysed after Cohen et al., (2017), and were explicitly presented in the discussion and methodology sections.

The following phrase was added on line 201 after “were calculated (see Methodology)”: “for each year studied.”

The second paragraph of the section of **Health and climate co-benefits of the rural residential energy transition** was rewritten as:

“This study found that the rural residential emission-associated premature deaths were as high as 570,000 (370,000-710,000 with the 95% uncertainty interval) in 1992 and decreased to 210,000 (140,000-260,000) in 2012 according to the GEMM model. Although baseline mortality rates changed rapidly during this period, the decrease in the mortality attributable to rural residential emissions was still significant from 1992 to 2012, with the influence of the population growth, aging, and age-standardised mortality rate adjusted (**Fig. S2**). The estimated number of deaths avoided due to decreased rural residential exposure was approximately 130,000 (90,000-160,000). During the same period, because of general increase in emissions from other anthropogenic sources, the total premature deaths induced by exposure to PM_{2.5} from all sources increased by 15%, which is consistent with the trend in the GBD study (*Cohen et al., 2017*). This outcome indicates the important health consequence of this emission source as well as the significant health benefits of the rural energy transition.”

Fig. S2 Contributions of population growth, population ageing, age-adjusted mortality rate, and exposure to the net change in the premature death from 1992 to 2012. The exposure was due to (A) residential emission, and (B) all sources.

The sentence on lines 11-13 in the Abstract was also revised as: “The change reduced 130,000 (90,000-160,000) mortality attributable to PM_{2.5} exposure, even though the premature deaths attributable to ambient PM_{2.5} from all sources increased during this 20-year period.”.

The following sentences were added to the section of in the Methodology: “The relative contributions of exposure, population growth, population ageing, and baseline mortality rates were analysed after Cohen *et al.*, (2017).”.

References

Cohen, A.J. et al., Estimates and 25-year trends of the global burden of disease attributable to ambient air pollution: an analysis of data from the Global Burden of Diseases Study 2015. *Lancet* 389, 1907-1918 (2017).

Comment

3) The importance of possible meteorological variability and possibly climate change on the results in the few years modeled is not discussed. I appreciate that it is likely infeasible to run each year individually over this 20-year period, but then impacts of variability and change on the results should be discussed. Was 2012 perhaps an unusually dry year with higher PM_{2.5} as a result?

Response

Because the energy survey was conducted for only the four study years, if the model is to be run for all 20 years, the emissions would have to be interpolated for all other years, which will introduce extra uncertainty. Additionally, even though the modelling was conducted for the entire period, the confounding effects of meteorological conditions cannot be distinguished by the modelling results. Fortunately, a reduced-form

model has been developed recently for quantitatively distinguishing influences of emissions from meteorological confounding effects by our group (Zhong *et al.*, 2018). The strategy was to model air quality for the three scenarios of 1) realistic emissions and meteorology, 2) realistic emissions but average meteorology (a typical year of 2014), and 3) average emissions and realistic meteorology. By analyzing the output of the three scenarios, reduced-form models were developed to predict emission associated air quality that eliminated meteorological confounding influences and probabilistic models were established to characterize meteorology associated fluctuation. One results from the study was the variation of meteorological influences on air quality over the 35 years from 1980 to 2014, covering the four years in this study, which is shown in Fig. 4 of the literature and borrowed here (above). It has been demonstrated that the meteorology induced variations from the long-term average were generally small for the four studied years of 2002, 2007, and 2012, with a single exception of 1992 (arrows in the figure).

A sentence was added to address the potential influence of meteorological fluctuation before “The model performance was previously evaluated ...” in the **Atmospheric Chemical Transport Modelling** section: “According to the results of a reduced-form modelling, the meteorological conditions in these four years were close to the multi-year average (Zhong *et al.*, 2018).”.

References

Zhong, Q., et al. Distinguishing emission-associated ambient air PM_{2.5} concentrations and meteorological factor-induced fluctuations. *Environ. Sci. Technol.* 52, 10416-10425 (2018).

Comment

4) I am unclear on the “downscaling” method used to get PM_{2.5} at very fine resolution, but the impacts of this method should be discussed further.

Response

To clarify the method, the sentence on line 315 was revised as: “In brief, the model-calculated concentrations were interpolated to a finer resolution using 0.1 degree emission inventory as a proxy with the wind-field justified. The detailed downscaling method can be found elsewhere (Shen et al., 2017). A commonly used method for downscaling is land use regression, in which land use is primarily used as a proxy for emissions (Hoek et al., 2008). It is reasonable to expect that direct use of emissions in this study can provide better results. A statistically significant relationship between emissions and annual mean PM_{2.5} concentrations in individual grids has been demonstrated (Zhong et al., 2018).”.

References

Shen, H. Z. et al. Urbanization-induced population migration has reduced ambient PM_{2.5} concentration in China. *Sci. Adv.* 3, e1700300 (2017).

Hoek, G. A review of land-use regression models to assess spatial variation of outdoor air pollution. *Atmos. Environ.* 42, 7561-7578 (2008).

Zhong, Q., et al. Distinguishing emission-associated ambient air PM_{2.5} concentrations and meteorological factor-induced fluctuations. *Environ. Sci. Technol.* 52, 10416-10425 (2018).

Comment

5) Please summarize the results of the evaluation of the model in the main text. I don't think it is sufficient to say that it has been “validated” without any more discussion in the main text. A NMB of 67% and NME of 91% for Black Carbon (Fig. S7) would not meet model performance criteria for the US. That may be the best that can be done currently in China, but the limitations of the model should be more apparent in the main text.

Response

The sentence on line 324 was replaced by the following ones: “Although the simulated results positively biased for total PM_{2.5} (NMB=25% and 30% for Beijing and Shanghai, respectively), black carbon (NMB=67%), nitrate (NMB=49%), and ammonium (NMB=38%), and negatively biased for particulate sulfate (NMB=-12%), the model predicted temporal and spatial variations of ambient PM_{2.5} are generally acceptable. The discrepancy is very likely from uncertainties in model inputs, missing mechanisms in the model (Tuccella et al., 2012; Wang et al., 2016), as well as high uncertainties in limited particle composition observational data with different sampling and analytical methods. The bias showed a spatial pattern with positive and negative values in eastern and western China, respectively, which are illustrated in **Fig. S9**. The overestimation in eastern China is attributed to the fact that the model was run for 2012, whereas observational data were collected in 2015 and a significant decrease has occurred since 2014. On the other hand, the influence of dust in the west is likely one important reason for the underestimation in this region.”.

Meanwhile, the detailed composition validation was revised to compare individual components separately and the results are provided in **Fig. S8**.

References

Tuccella, P.; Curci, G.; Visconti, G.; Bessagnet, B.; Menut, L.; Park, R. Modelling of gas and aerosol with WRF/Chem over Europe: evaluation and sensitivity study. *J. Geophys. Res.* 117, D03303 (2012)

Wang, L.; Zhang, Y.; Wang, K.; Zheng, B.; Zhang, Q.; and Wei, W. Application of Weather Research and Forecasting Model with Chemistry (WRF/Chem) over northern China: sensitivity study, comparative evaluation, and policy implication. *Atmos. Environ.* 124, 337-350 (2016).

Zheng, B. et al., Heterogeneous chemistry: a mechanism missing in current models to explain secondary inorganic aerosol formation during the January 2013 haze episode in North China. *Atmos. Chem. Phys.* 15, 2031-2049 (2015).

Fig. S8 Comparison of the modelled concentrations of major $PM_{2.5}$ components including black carbon (A), organic carbon (B), sulfate (C), nitrate (D), and ammonium (E) with those observed. The dashed and dotted lines represent the error ranges of two and five times, respectively. Sample size (N), normalized mean bias (NMB) and normalized mean error (NME) are also shown. Source data are provided as a Source Data file.

Fig. S9 Distribution of the relative difference in the model calculated $PM_{2.5}$ concentrations in 2012 in comparison with the observation in 2015. Source data are provided as a Source Data file.

Comment

6) 1.330 – I'd like to see the “proportional method” described more with an equation (could be in the supplement).

Response

The sentence was revised as “To quantitatively evaluate the contributions from rural residential sources, a normalized marginal method (Supplement S1) (UNFCCC, 2002; Trudinger and Enting, 2005) was adopted by running the model three times using emission scenarios of ...”

A detailed description of this method was added to the supplement:

Supplement S1: Normalized marginal method

Due to the nonlinear relationship between the emissions and the modelled PM_{2.5} concentrations, a normalized marginal method was applied to calculate the relative contributions of rural residential sources to ambient PM_{2.5} concentrations.

We performed the simulations using three emission scenarios of: 1) a 20% reduction in the rural residential sources (r-20%); 2) a 20% reduction in all but the residential sources (o-20%); and 3) all sources. We assessed the relative contributions of rural residential sources for each simulated grid using the following equation: $RC_r = (C_{all} - C_{r-20\%}) / (2 \times C_{all} - C_{r-20\%} - C_{o-20\%})$, where RC_r is the relative contribution of rural residential emissions to ambient PM_{2.5}; C_{all} , $C_{r-20\%}$ and $C_{o-20\%}$ are the ambient PM_{2.5} concentrations in the three scenarios of all sources, a 20% reduction in the rural residential sources, and a 20% reduction in all but the residential sources, respectively. The absolute contribution of rural residential emissions was calculated as $RC_r \times C_{all}$ for each grid.

The OSCAR model was run to assess the contribution of rural residential emissions to radiative forcing. The model was run three times with the three scenarios, the same as that in the chemical transport modelling, and the relative contribution was calculated as $(RF_{all} - RF_{r-20\%}) / (2 \times RF_{all} - RF_{r-20\%} - RF_{o-20\%})$, where RF_{all} , $RF_{r-20\%}$ and $RF_{o-20\%}$ are the radiative forcing values in the three scenarios of all sources, a 20% reduction in the rural residential sources, and a 20% reduction in all but the residential sources, respectively.

Comment

7) 1. 341 – The GEMM model is used for the mortality calculations. This is new and is not used in standard practice, such as by GBD, although perhaps it will become in more widespread use soon. GEMM should give many more deaths than the recent alternatives. The authors are free to pick the function they like, but they should explain why they chose GEMM over alternatives and describe the consequences of this choice. They might also consider a sensitivity analysis with another widely used function.

Response

We used the GBD method before and was suggested by another reviewer to update the dose-response relationship during the first round of review. Therefore, the new GEMM model was applied in the revised version. According to the comment, we compared the results of the two models and the result was added to the latest version.

The following sentences were added on line 347: “Health impact models, especially does-response functions, affect estimated premature deaths considerably. For comparison, premature deaths were also calculated using the Integrated Risk Function developed by GBD (Cohen et al., 2017) and the results are shown and compared in the Supplement **Table S3**. As expected, the GEMM model yielded more premature deaths than the GBD model. However, the relative contributions of rural residential emissions to the total premature deaths and the temporal trend were similar between these two approaches (**Fig. S3**).”.

A table (Table S3) was added to the supplement:

Table S3. for Estimated premature deaths (medium with 95% uncertainty interval) associated with the five diseases (ALRI, LC, IHD, STROKE, and COPD) attributable to rural residential emissions for the

four study years following the GEMM and GBD models.

Year		GEMM	GBD
1992	ALRI	77,000 (46,000-98,000)	40,000 (30,000-50,000)
	LC	34,000 (22,000-43,000)	20,000 (10,000-30,000)
	IHD	150,000 (140,000-160,000)	80,000 (30,000-120,000)
	STROKE	200,000 (110,000-260,000)	110,000 (60,000-180,000)
	COPD	110,000 (60,000-150,000)	90,000 (50,000-130,000)
	TOTAL	570,000 (370,000-710,000)	340,000 (200,000-500,000)
2002	ALRI	33,000 (20,000-42,000)	20,000 (10,000-30,000)
	LC	36,000 (24,000-45,000)	20,000 (10,000-30,000)
	IHD	130,000 (120,000-140,000)	70,000 (40,000-100,000)
	STROKE	160,000 (90,000-220,000)	90,000 (50,000-140,000)
	COPD	82,000 (44,000-110,000)	70,000 (40,000-100,000)
	TOTAL	440,000 (300,000-560,000)	270,000 (150,000-390,000)
2007	ALRI	17,000 (10,000-21,000)	9,000 (7,000-20,000)
	LC	30,000 (20,000-38,000)	20,000 (10,000-30,000)
	IHD	94,000 (89,000-100,000)	50,000 (20,000-70,000)
	STROKE	110,000 (58,000-140,000)	60,000 (30,000-90,000)
	COPD	50,000 (27,000-67,000)	40,000 (30,000-60,000)
	TOTAL	300,000 (200,000-360,000)	170,000 (100,000-250,000)
2012	ALRI	12,000 (7,000-15,000)	7,000 (5,000-9,000)
	LC	24,000 (16,000-30,000)	15,000 (10,000-20,000)
	IHD	67,000 (62,000-70,000)	30,000 (20,000-50,000)
	STROKE	70,000 (39,000-93,000)	40,000 (20,000-50,000)
	COPD	36,000 (19,000-48,000)	30,000 (20,000-40,000)
	TOTAL	210,000 (140,000-260,000)	120,000 (70,000-180,000)

Meantime, **Figure S3** was replaced with the following one:

Fig. S3 Temporal trends of relative contributions of rural residential emissions to total premature deaths induced by exposure to ambient air PM_{2.5} in China from 1992 to 2012, estimated following the GEMM and GBD approaches. Source data are provided as a Source Data file.

A phrase “from the GEMM model” was added on line 207.

References

Cohen, A.J. et al., Estimates and 25-year trends of the global burden of disease attributable to ambient air pollution: an analysis of data from the Global Burden of Diseases Study 2015. *Lancet* 389, 1907-1918 (2017).

Comment

8) 1. 348-360 – More details on the radiative forcing calculations are provided in this draft, but the description does not do a good job yet of giving an understanding of how the calculations are done, such as what OSCAR is used for and what the marginal method does. Just stating inputs and outputs of each step would be an improvement. The calculation leaves out contributions from ozone and methane forcing from changes in emissions of CO, NO_x and VOCs. That omission might be justified, but the authors could discuss in the context of other studies that have estimated these quantities for residential emissions or for biomass burning generally.

Response

The sentence on line 330 was revised as “To quantitatively evaluate the contributions from rural residential sources, a normalized marginal method (Supplement S1) (UNFCCC, 2002; Trudinger and Enting, 2005) was adopted ...”, and the following paragraph was added to the Supplement to provide more details in the calculation.

“The OSCAR model was run to assess the contribution of rural residential emissions to radiative forcing. The model was run three times with the three scenarios, the same as that in the chemical transport modelling, and the relative contribution was calculated as $(RF_{all}-RF_{r-20\%})/(2\times RF_{all}-RF_{r-20\%}-RF_{o-20\%})$, where RF_{all} , $RF_{r-20\%}$ and $RF_{o-20\%}$ are the radiative forcing values in the three scenarios of all sources, a 20% reduction in the rural residential sources, and a 20% reduction in all but the residential sources, respectively.”

The sentences on lines 350-355 were further revised to provide OSCAR model details: “To estimate the radiative forcing, the global biogeochemical cycle model OSCAR (v2.1) was used (Li et al., 2016). The OSCAR model is a reduced-form “Earth system change” model used widely in the climate change research community. This model was developed with the three principles of embedding as many components and processes as possible, building as a meta-model capable of emulating the sensitivity of models with higher resolution or superior complexity, and comprising as a dynamic model of the Earth system (Gasser et al., 2017). The model outputs include radiative forcing of each climate-relevant component and global surface temperature change, and the inputs are emissions of various compounds and drivers related to land-use and land-cover change. A detailed description of the model was presented by Gasser *et al.* (2017). To estimate the contributions from rural residential sources, the model was run three times using different emission scenarios, in line with those model scenarios in the atmospheric chemical transport modelling, and the normalized marginal method was used (S1).”

The sentence on line 356 was revised as: “The derived climate forcing components discussed in this study included CO₂, BC, POA, sulfate and nitrate, of which residential coal and biomass burning are usually the larger emitters of these compounds or their precursors. The contributions of residential combustion emissions to ozone precursors and methane are relatively small compared to its contribution to aerosol (Li et al., 2017; Peng et al., 2016).”

The sentence on line 357 was deleted.

References

- Gasser, T., Ciais, P., Boucher, O., Quilcaille, Y., Tortora, M., Bopp, L., & Hauglustaine, D. The compact Earth system model OSCAR v2.2: description and first results. *Geosci. Model Dev.* 10, 271-319 (2017).
- Li, B., et al., The contribution of China’s emissions to global climate forcing. *Nature*, 531, 357-361 (2016).
- Li, M. et al., MIX a mosaic Asian anthropogenic emission inventory under the international collaboration framework of the MICS-Asia and HTAP. *Atmos. Chem. Phys.* 17, 935-963 (2017).
- Peng, S. et al. Inventory of anthropogenic methane emissions in mainland China from 1980 to 2010. *Atmos. Chem. Phys.* 16, 14545-14562 (2016).
- Trudinger, C. & Enting, I. Comparison of formalisms for attributing responsibility for climate change:

non-linearities in the Brazilian proposal approach. *Climatic Change* 68, 67-99 (2005).
United Nations Framework Convention on Climate Change (UNFCCC). Methodological Issues: Scientific and Methodological Assessment of Contributions to Climate Change, Report of the Expert Meeting, Note by the Secretariat. <http://unfccc.int/resource/docs/2002/sbsta/inf14.pdf> (UNFCCC, 2002).

Comment

9) 1. 367 – the authors assume a 10% uncertainty in PM_{2.5}. I don't think that the modeled PM_{2.5} is likely within 10% of reality, nor do I think we know PM_{2.5} from rural residential sources within 10%. - Elsewhere in the paper (e.g., l. 16), concentrations are shown with an uncertainty that is much greater than 10%. This discrepancy is not explained. It could be that the authors are considering variability among grid cells in the wider uncertainty range shown for concentration results, but that is not explained.

Response

This point was not explained clearly. The 10% variability is assumed for the PM_{2.5} in each grid, and this value was then used in the uncertainty analysis for health impacts. The modelled PM_{2.5} had much larger overall uncertainty, which is mainly due to spatial variability among grids. According to these comments, we recalculated and compared the uncertainty of the calculated premature deaths by assuming 10%, 15%, and 25% variations of PM_{2.5} concentrations in individual grid cells, respectively. The results are presented in Figure S10 as 95% confidence intervals.

The sentence on line 367-368 was revised as “Therefore, for each grid, a 10% uncertainty in PM_{2.5} was assumed and used in the uncertainty analysis for the health impact assessment. The sensitivity analysis (Figure S10) showed that differences in estimated premature deaths with the assumed grid PM_{2.5} concentration uncertainties at 10%, 15% and 25% were within 1%, indicating that the overall uncertainty in health outcomes was affected largely by dose-response functions compared with the uncertainty in PM_{2.5} concentration (Liu et al., 2016).”.

Fig. S10 Estimated premature deaths (medium with 95% uncertainty interval) associated with the five diseases (ALRI, LC, IHD, STROKE, and COPD) attributable to rural residential emissions in 2012. The uncertainty intervals were derived from uncertainties in parameters in the dose-exposure relationship as well as the PM_{2.5} exposure level in which 10%, 15%, and 25% variations of PM_{2.5} concentration at individual grid cells were assumed and compared.

References

Liu, J., Han, Y., Tang, X., Zhu, J., & Zhu, T. Estimating adult mortality attributable to PM_{2.5} exposure in China with assimilated PM_{2.5} concentrations based on a ground monitoring network. *Sci. Total Environ.* 568, 1253-1262 (2017).

Comment

10) Fig 2. – I think my earlier comments still hold. If this is shown as illustration, then say so, but it

leaves the question open as to whether the rest of the year is like this one month in one province. More broadly, I'm not sure why daily variations (while they are interesting) are important for this paper.

Response

The figure and associated text were deleted.

Comment

11) The authors explain how they calculate PWC PM_{2.5} in the response to author document, but I don't see where that appears in the current draft. And it is not intuitive. I understand what a population-weighted average concentration over many grid cells is $\text{Sum}(\text{Pop}_i \cdot \text{Conc}_i) / \text{Sum}(\text{Pop}_i)$ and the authors use that in some places. But they also use PWC for a single grid cell which they define (in the response to my earlier comments) as $(\text{Pop} \cdot \text{Conc}) / (\text{Average Pop of all grid cells})$. I see that there is value in this quantity as a way of representing how important high concentrations are for population exposure. But it is also not very clear what the physical interpretation of this quantity is. Figure 3b have values approaching 500 $\mu\text{g m}^{-3}$ – what does that mean? Definitions should appear in the paper.

Response

The definition of PWC was added on line 337 in the Method section: “The PWC of a region with more than one grid was calculated as $\Sigma(P_i \cdot C_i) / P_t$, where P_i and C_i are population and air PM_{2.5} concentration at the i^{th} grid, and P_t is the total population in the domain of interest.”.

Thank you for pointing out the problem with **Figure 3**, in which an intermediate variable with no physical meaning was used, making the figure difficult to understand. In the revised version, PWC values of all provinces in mainland China were calculated and mapped. The results are compared with a map showing ambient air PM_{2.5} concentrations.

Fig. 2 Provincial mean air quality concentrations (A) and population-weighted PM_{2.5} concentrations (B) attributable to rural residential emissions in 2012 in mainland China. Source data are provided as a Source Data file.

The first sentence of the section “**Strong spatial variations with higher contributions in eastern China**” was revised as “In addition to the temporal changes, there was also strong spatial variation in the contributions of rural residential emissions to ambient PM_{2.5} concentrations, which is shown in **Fig. 2A** as provincial averages.”. The last two sentences of the paragraph were revised as: “The spatial variation was enhanced by population weighting for assessing exposure, as shown by provincial averages in **Fig. 2B**. The overlay of the sources (emissions) and receptors (population) led to an even sharper contrast between the heavily and less polluted regions”.

Comment

12) 1.78 – are you saying that the PWC for residential emissions is 3x that for PM from all sources? That seems counter-intuitive, and speaks to the interpretability of PWC as an indicator.

Response

This point was not clearly presented. The PWC associated with residential emissions is three times the ambient PM_{2.5} emitted from residential sources. The sentence was revised accordingly: “The national average rural residential source associated PWC (14 µg/m³) was almost three times the average ambient PM_{2.5} concentration (5.4 µg/m³) attributable to rural residential emissions.”

Reviewers' comments:

Reviewer #1 (Remarks to the Author):

Questions are well addressed. I don't have any further question.

Reviewer #3 (Remarks to the Author):

Summary

Thanks to the authors for revising the manuscript and for providing evaluation of the meteorological model predictions for the years modeled.

However, my concern with the air quality model evaluation has not been satisfactorily addressed. In fact, I see more problems that have cropped up in this revision.

Lines 332-335: In presenting the air quality model evaluation, the authors state "The overestimation in eastern China is attributed to the fact that the model was run for 2012, whereas observational data were collected in 2015 and a significant decrease has occurred since 2014."

I find it surprising that there no observations available in China for the modelled years other than in Beijing and Shanghai for 2012. I don't see a scientific justification for this 3-year offset for the air quality model evaluation, other than that of inconvenience (lack of access to observations to the authors). I am not convinced about the robustness of the findings related to the contribution of residential combustion to air quality and health risk during different time periods in rural vs. urban China without having confidence in the baseline air quality model predictions. The entire discussion on model evaluation presented in Lines 321 – 332 is less meaningful, when the authors conclude that section by pointing out the 3-year offset.

Figure S7: The caption states "Since the calculations and observations were for different years, the concentrations were standardized.". Please clarify what exactly was done to standardize. Were these for the observed or modeled, or both?

Figure S9: Has no useful purpose given the 3-year offset. What has significantly decreased since 2014 and where? Any citable references for this?

Response to comments

NCOMMS-18-25418B

Title: Air Pollution from Rural Chinese Households under Fast Residential Energy Transition

Reviewer-1

Comment

Questions are well addressed. I don't have any further question.

Response

Thank you again for reviewing and valuable comments that did help us to improve the manuscript.

Reviewer-3

Comment

Summary-Thanks to the authors for revising the manuscript and for providing evaluation of the meteorological model predictions for the years modeled.

Response

It is our great pleasure to do this. Thank you again for the important comments and suggestions on the clarification and evaluation of the meteorological model and inputs.

Comment

However, my concern with the air quality model evaluation has not been satisfactorily addressed. In fact, I see more problems that have cropped up in this revision.

Lines 332-335: In presenting the air quality model evaluation, the authors state “The overestimation in eastern China is attributed to the fact that the model was run for 2012, whereas observational data were collected in 2015 and a significant decrease has occurred since 2014.”

I find it surprising that there no observations available in China for the modelled years other than in Beijing and Shanghai for 2012. I don't see a scientific justification for this 3-year offset for the air quality model evaluation, other than that of inconvenience (lack of access to observations to the authors). I am not convinced about the robustness of the findings related to the contribution of residential combustion to air quality and health risk during different time periods in rural vs. urban China without having confidence in the baseline air quality model predictions. The entire discussion on model evaluation presented in Lines 321 – 332 is less meaningful, when the authors conclude that section by pointing out the 3-year offset.

Response

Official routine PM_{2.5} monitoring program of China started in 2013 (Batterman et al., 2016; Zhang and Cao, 2015). That was the reason we chose to validate our model using the monitoring data in 2015 to evaluate the modelled PM_{2.5} spatial distribution. We agree with the reviewer that such an offset can lead to high uncertainties. Following the reviewer's comment, we tried our best to collect as many as possible observation data reported in the literature (Geng et al., 2015; Wang et al., 2017) to validate the model in the revised version. Efforts are also made to compare the modelled concentrations and the spatial pattern with those retrieved from validated and widely used satellite remote sensing derived ground level PM_{2.5} concentrations based Aerosol Optical Depth (AOD) by different groups in the scientific community (Ma et al., 2016; van Donkelaar et al., 2015) and PM_{2.5} estimated from the historical visibility record in China (Liu et al., 2017). Overall, our modelled PM_{2.5} levels and spatial pattern agree reasonably well with the literature reported and satellite derived data.

The sentences from line 321 to 336 were then rewritten as follows accordingly:

“Prior to 2013, there was no official routine monitoring program for ambient PM_{2.5} in China (*Geng et al., 2015; Wang et al., 2017*), and sampled PM_{2.5} data were only available from several field studies reported in the literature and those released from the U.S. Embassy. To validate our modelling results, in addition to comparing with available observational data, the modelled PM_{2.5} concentrations were also compared to those retrieved from the satellite remote sensing which have been widely used in the scientific community (*Ma et al., 2016; van Donkelaar et al., 2015; Martin, 2019*), and those estimated based on visibility records in China (*Liu et al., 2017*). Overall, our model simulated concentrations were slightly higher than the observation data (NMB=25-30%, **Fig. S6** and NMB=3%, **Fig. S7**), but considerably lower than those retrieved from the satellite and visibility retrieved results (NMB= -61% ~ -5%, **Table S3**). **Fig. S8** shows the spatial pattern of the differences between the modelled PM_{2.5} concentrations in this study and those retrieved from the satellite derived data (*Ma et al., 2016*). Our results were lower in the sparsely populated western area and higher in some southeast sites. Previous studies found that satellite-derived PM_{2.5} might underestimate PM_{2.5} levels during heavy pollution episodes and overestimate PM_{2.5} concentrations in low pollution cases (*Ma et al., 2014; Geng et al., 2015; Guo et al., 2016*). The discrepancy is also likely from uncertainties in chemical transport model inputs such as air emissions of PM_{2.5} precursors, and missing mechanisms in the model (*Tuccella et al., 2012; Wang et al., 2016; Zheng et al., 2015*). We further compared the modelled major PM_{2.5} components with measured data, including black carbon (NMB=67%), organic carbon (NMB=-3%), sulfate (NMB=-12%), nitrate (NMB=49%), and ammonium (NMB=38%) (**Fig. S9**). The differences could come from the uncertainties in modelling, as well as uncertainties in limited observations with different sampling and analytical methods. With all the results taken into consideration, the model-predicted ambient PM_{2.5} concentrations and the spatiotemporal variations are generally acceptable.”

Fig. S7 Comparison between the model-calculated PM_{2.5} concentrations and those from field measurements reported in the literature. The dashed and dotted lines represent the error ranges of two and five times, respectively. Normalized mean bias (NMB) and normalized mean error (NME) are also shown. Source data are provided as a Source Data file.

	Modelled PM _{2.5} (this study) vs. AOD-PM _{2.5} (CTM, van Donkelaar et al., 2015)			Modelled PM _{2.5} (this study) vs. AOD-PM _{2.5} (statistical model, Ma et al., 2016)			Modelled PM _{2.5} (this study) vs. visibility-based PM _{2.5} (Liu et al., 2017)		
	N	NMB	NME	N	NMB	NME	N	NMB	NME
1992	/	/	/	/	/	/	614	-45%	56%
2002	94294	-43%	50%	/	/	/	615	-51%	57%

2007	94294	-44%	51%	87628	-61%	63%	396	-32%	53%
2012	94294	0%	65%	84789	-35%	46%	378	-5%	46%

Table S3. Statistics for evaluating the modelled ambient PM_{2.5} against the estimated historical PM_{2.5} in past studies based on satellite-based Aerosol Optical Depth (AOD) using a chemical transport model (CTM) (van Donkelaar et al., 2015), from AOD with a statistical model (Ma et al., 2016), and that retrieved from the historical visibility records (Liu et al., 2017).

Fig. S8 Distribution of the relative difference between the model-calculated PM_{2.5} concentrations and satellite retrieved PM_{2.5} in 2012 (modelled annual average minus satellite retrieved concentrations) (Ma et al., 2016). Source data are provided in a Source Data file.

References

1. Wang, J. et al. Particulate matter pollution over China and the effects of control policies. *Sci. Total Environ.* 584-585, 426-447 (2017).
2. Batterman, S., Xu, L., Chen, F., Chen, F., & Zhong, X. Characteristics of PM_{2.5} concentrations across Beijing during 2013-2015. *Atmos. Environ.* 145, 104-114 (2016).
3. Zhang, Y., & Cao, F. Fine particulate matter (PM_{2.5}) in China at a city level. *Scientific Reports* 5, 14884 (2015).
4. Geng, G. et al. Estimating long-term PM_{2.5} concentrations in China using satellite-based aerosol optical depth and a chemical transport model. *Remote Sens. Environ.* 166, 262-270 (2015).
5. Ma, Z., Hu, W., Huang, L., Bi, J., Liu, Y. Estimating ground-level PM_{2.5} in China using satellite remote sensing. *Environ. Sci. Technol.* 48, 7436-7444 (2014).
6. Ma, Z. et al. Satellite-based spatiotemporal trends in PM_{2.5} concentrations in China, 2004-2013. *Environ. Health Perspect.* 124, 184-192 (2016).
7. Liu, M., Bi, J., Ma, Z. Visibility-based PM_{2.5} concentrations in China: 1957-1964 and 1973-2014. *Environ. Sci. Technol.* 51, 13161-13169 (2017).
8. van Donkelaar, A. et al. Global fine particulate matter concentrations from satellite for long-term exposure assessment. *Environ. Health Perspect.* 123, 135-143 (2015).
9. Martin, R. Atmospheric composition Analysis Groups. http://fizz.phys.dal.ca/~atmos/martin/?page_id=140. Accessed Jun. 2019.
10. Tuccella, P.; Curci, G.; Visconti, G.; Bessagnet, B.; Menut, L.; Park, R. Modelling of gas and aerosol with WRF/Chem over Europe: evaluation and sensitivity study. *J. Geophys. Res.* 117, D03303 (2012)
11. Wang, L.; Zhang, Y.; Wang, K.; Zheng, B.; Zhang, Q.; and Wei, W. Application of Weather Research and Forecasting Model with Chemistry (WRF/Chem) over northern China: sensitivity study, comparative evaluation, and policy implication. *Atmos. Environ.* 124, 337-350 (2016).
12. Zheng, B. et al., Heterogeneous chemistry: a mechanism missing in current models to explain secondary inorganic aerosol formation during the January 2013 haze episode in North China. *Atmos. Chem. Phys.* 15, 2031-2049 (2015).
13. Guo, H., et al. Comparison of four ground-level PM_{2.5} estimation models using PARASOL aerosol optical depth data from China. *Inter. J. Environ. Res. Public Health* 13, 180 (2016)

Comment

Figure S7: The caption states “Since the calculations and observations were for different years, the concentrations were standardized.”. Please clarify what exactly was done to standardize. Were these for the observed or modeled, or both?

Response

The model validation was totally updated using data in the same year and the old figure S7 together with the sentences in previous version was now deleted.

Comment

Figure S9: Has no useful purpose given the 3-year offset. What has significantly decreased since 2014 and where? Any citable references for this?

Response

As mentioned in the response above, we reconducted model validation and rewrote the section. This figure and the sentences on lines 332-336 were removed.

REVIEWERS' COMMENTS:

Reviewer #3 (Remarks to the Author):

Summary

Thanks to the authors for revising the manuscript and for providing evaluation of the air quality model predictions for the years modeled, using data from prior publications and satellite retrievals.

Minor comments:

Figures S7 and S9: I don't see why these use log-scale. Ranges of PM_{2.5} and speciated concentrations from observed and modeled studies are usually and better compared using linear scales. Also, if you provide scatter plots, it is useful to present correlation coefficients as well.

Lines 416 – 417 and Figure S8 Caption: The authors state "Data availability. The source data underlying Figs. 1-4, supplementary Figs. S1-S4 and S6-S10 and supplementary Tables S1-S4 are provided as a Source Data file.

- Data for Fig S8 not included in the provided Excel file. Please include as stated.
- Data for Tables S1-S4 are also not included. Please include as stated.

Response to referees' comments

NCOMMS-18-25418C

Title: Air Pollution from Rural Chinese Households under Fast Residential Energy Transition

Authors: Shen *et al.*,

Reviewer-3

Comment

Summary -Thanks to the authors for revising the manuscript and for providing evaluation of the air quality model predictions for the years modeled, using data from prior publications and satellite retrievals.

Minor comments:

Figures S7 and S9: I don't see why these use log-scale. Ranges of PM2.5 and speciated concentrations from observed and modeled studies are usually and better compared using linear scales. Also, if you provide scatter plots, it is useful to present correlation coefficients as well.

Response

These figures are revised using linear scales, and correlation coefficients are added accordingly.

Comment

Lines 416 – 417 and Figure S8 Caption: The authors state “Data availability. The source data underlying Figs. 1-4, supplementary Figs. S1-S4 and S6-S10 and supplementary Tables S1-S4 are provided as a Source Data file.

- Data for Fig S8 not included in the provided Excel file. Please include as stated.
- Data for Tables S1-S4 are also not included. Please include as stated.

Response

Added.